# Johnson-Lindenstrauss Transforms in Distributed Optimization

## Abstract

Increasing volumes of data and models in the machine learning demand efficient methods. Distributed optimization addresses these challenges, for instance, by utilizing compression mechanisms, that reduce the number of bits transmitted. One of the known techniques, that diminish the dimension of the database are Johnson-Lindenstrauss (JL) mappings, that benefit from the ease of implementation. Unlike the usual sparsification techniques, they preserve the scalar product and distances between the vectors, which is beneficial for advanced distributed machine learning problems, such as byzantine-robust learning, personalized and vertical federated learning. In this paper, we close the gap and connect JL Transforms with optimization algorithms and demonstrate, that we can compress communication messages with them. We also validate our theoretical results by the conducted experiments.

## 1 Introduction

In recent years, the increasing scale of data and computational demands in machine learning (Kairouz et al., 2021) has necessitated the development of distributed optimization (Tang et al., 2020), which tries to minimize the global model:

$$\min_{x \in \mathbb{R}^n} \left\{ f(x) = \frac{1}{M} \sum_{i=1}^{M} f_i(x) \right\}, \tag{1}$$

$$\text{with } f_i(x) = \frac{1}{n_i} \sum_{j=1}^{n_i} l(g(x, a_i^j), b_i^j),$$

where $f_i$ stands for the loss of the device $i$, depending only in its local data $\left\{ a_i^j, b_i^j \right\}_{j=1}^{n_i}$, $n_i$ for local dataset size and $g$ for the model with parameters $x$. Employed methods enable efficient training across multiple nodes, reducing computation time and handling large-scale problems that are infeasible for single-machine processing. However, distributed optimization introduces challenges such as communication bottlenecks, synchronization overhead, and bandwidth limitations, which can significantly hinder performance (Li et al., 2019).

To overcome the issues of high communication costs in distributed optimization, various compression techniques have emerged as effective solutions (McMahan et al., 2017; Alistarh et al., 2017). These methods reduce the volume of transmitted data while retaining essential information, enabling efficient training without significantly hindering convergence. Common approaches include quantization (Yu et al., 2019), which reduces numerical precision and sparsification (Wangni et al., 2018), which transmits only the most significant components. The choice of communication strategy depends on the specific distributed learning framework (Vanhaesebrouck et al., 2017). In classical distributed optimization, workers may transmit either compressed gradients (Alistarh et al., 2017) or some sequences, based on gradients, proposed to correct compression artifacts over the iterations (Karimireddy et al., 2019; Richtárik et al., 2021), (Mishchenko et al., 2024). In both cases, final gradient estimation, that is used to perform the optimization step, is an average of some compressed vectors.

However, several setups are based on interconnections between the gradients, rather than the mean. For instance, in Byzantine-robust optimization (Karimireddy et al., 2021), where workers can be categorized as either honest or Byzantine, attempting to disrupt the convergence process by sending

poison messages. One of the approaches to overcome this difficulty is to assign coefficients to clients based on their reliability (Cao et al., 2020), using these trust scores to perform gradient steps. This can be done by comparing the scalar product between the gradient of the whole model and the local ones (Yan et al., 2023; Molodtsov et al., 2025). The motivation is to exclude the clients, whose gradients are not co-directed with the average one. Hence, the step is the following:

$$
\omega_i^t = \frac{\max\left\{\langle \nabla f_i(x^t), \nabla f_0(x^t)\rangle, 0\right\}}{\sum\limits_{j=1}^{M} \max\left\{\langle \nabla f_j(x^t), \nabla f_0(x^t)\rangle, 0\right\}}, \tag{2}
$$

$$
x^{t+1} = x^t - \gamma \sum_{i=1}^{M} \omega_i^t \nabla f_i(x^t),
$$

where $f_0$ is a function, located on a trusted device. Therefore, if we add the compression in the byzantine setting, it should preserve the scalar products to maintain the reasonable assigning trust scores to the clients.

Another sphere of distributed optimization implementation is federated learning (FL) (Wen et al., 2023). In horizontal (personalized) FL, model is trained with similar features on multiple computing nodes, thus, basic approaches are equivalent to the problem (1) (McMahan et al., 2017). However, as this setup is characterized with highly heterogeneous and spread across the devices individual data, more advanced methods can be interpreted as an ensemble of personalized local models (Smith et al., 2017; Hanzely et al., 2020), (Hanzely & Richtárik, 2020). In these approaches each participating device updates its weights based not only on its own data, but also on interactions with other clients. This can be formalized by penalizing the deviations of the local models' states:

$$
\min_{x_1,\ldots,x_M \in \mathbb{R}^n} \left\{ f(x_1,\ldots,x_M) = \frac{1}{M}\sum_{m=1}^{M} f_m(x_m) + \frac{\lambda}{2M}\sum_{m=1}^{M} \|x_m - \overline{x}\|^2 \right\}, \tag{3}
$$

where $\overline{x}$ is the average of parameters $x_1,\ldots,x_M$, which are stored on nodes locally. Though, the first term can be minimized locally, to optimize the second one we need to communicate and compute $\overline{x}$. Hence, if we speak about reducing the communication cost, we need not to perturb the $l_2$-norm between the local states $x_m$ and the global one $\overline{x}$ when we use compressed messages.

Besides horizontal FL, there is a vertical one (Liu et al., 2024), where the data is split across features, instead of the samples, as in horizontal. Particularly, in the horizontal setup, we split the feature matrix $A$ by rows and end up with problems (1) and (3). In vertical FL, we split it by columns – $A = [A_1,\ldots,A_M]$ with $A_i \in \mathbb{R}^{m \times n_i}$ and $\sum_i n_i = n$. For the ease of notation let us consider equation linear model in vertical setting:

$$
\min_{x_i \in \mathbb{R}^{n_i}} l\left(\sum_{i=1}^{M} A_i x_i;\ b\right) + \sum_{i=1}^{M} r_i(x_i),
$$

with the parameter vector $x$ is $x = (x_1,\ldots,x_M)$ with $x_i \in \mathbb{R}^{n_i}$. We assume regularization function $r$ to be separable – $r(x) = \sum_i r_i(x_i)$.

This problem can be rewritten with introducing new variables:

$$
\min_{x_i, z_i \in \mathbb{R}^n} l\left(\sum_{i=1}^{M} z_i;\ b\right) + \sum_{i=1}^{M} r_i(z_i) \tag{4}
$$

$$
\text{s.t.} \qquad A_i x_i - z_i = 0.
$$

Distributed Alternating Direction Method of Multipliers (ADMM) is a popular solver for these problems due to its ability to decompose the optimization problem into subproblems that can be solved locally with global consensus achieved by iterative updates. It was thoroughly investigated in (Boyd et al., 2011), where a compound view on various distributional problem states was given. Moreover, vertical FL scenario with ADMM solver was studied in (Xie et al., 2024) and (Samra et al., 2024).

Update scheme can be written as:

$$x_i^{t+1} = \underset{y \in \mathbb{R}^{n_i}}{\arg\min} \left\{ r_i(y) + (\rho/2) \left\| A_i(y - x_i^t) - \overline{z}^t + \overline{Ax}^t + u^t \right\|_2^2 \right\}, \tag{5}$$

$$\overline{z}^{t+1} = \underset{z \in \mathbb{R}^m}{\arg\min} \left\{ l(M\overline{z} - b) + (M\rho/2) \left\| z - \overline{Ax}^{t+1} - u^t \right\|_2^2 \right\}, \tag{6}$$

$$u^{t+1} = u^t + \overline{Ax}^{t+1} - \overline{z}^{t+1}, \tag{7}$$

where $\overline{Ax}^t = \frac{1}{M} \sum_{i=1}^{M} A_i x_i^t$. At every iteration each device sends $A_i x_i^t$ to a server, and receives $\overline{Ax}^t + u^t - \overline{z}^t$. Lines (5), which is conducted on local devices, and (6), which is done on server, can be regarded as minimization problems with a square regularization term. Hence, if we compress the messages between the server and the nodes, we need not preserve difference in $l_2$-norm between appropriate vectors.

Interestingly, highlighted methods rely on preserving the $l_2$-norm similarity between vectors rather than their exact values. Therefore, we might investigate compression operators, that do not disturb the $l_2$-norm. For instance, Johnson-Lindenstrauss (JL) Transforms (Johnson et al., 1984) are effective, since they project high-dimensional vectors into a lower-dimensional space while approximately preserving distances and scalar products. There is a deterministic way to compute this transformation, however, we focus on a stochastic variation, moreover, when this transform is a matrix, which elements are independent random variables. This is an effortless way to reduce the problem's dimension.

Dimensionality reduction techniques, particularly Johnson-Lindenstrauss Transforms, have significantly enhanced the efficiency of various modern applications. These mappings improve performance in nearest-neighbour search (Kushilevitz et al., 1998), (Andoni et al., 2015). Also, their distributional properties can make random partitioning (Kleinberg, 1997) more effective than basic resizing. They are also valuable in linear regression (Heinze et al., 2016; Thanei et al., 2017), enabling approximate solutions with reduced computational and time costs. Additionally, JL Transforms are helpful in clustering tasks (Tang et al., 2005; Boutsidis et al., 2010), including Gaussian mixtures (Dasgupta, 1999; Urruty et al., 2007), subspace clustering (Heckel et al., 2017), and graph partitioning (Guo et al., 2020). The inherent randomness in distributional JL Transforms facilitates differentially private algorithms (Upadhyay, 2014; Sheffet, 2017), safeguarding training data. Furthermore, they have demonstrated effectiveness in numerical linear algebra (Woodruff et al., 2014), such as low-rank approximations (Musco & Musco, 2020) and approximate matrix multiplication (Sarlos, 2006).

Though, JL Transforms are well-known for the scientific community, their applications to the distributed optimization and federative learning were lacking several vital properties. Mostly, they were considered as unbiased sketching mechanisms (Song et al., 2021; Shrivastava et al., 2024; Dai et al., 2019), where convergence in expectation was derived. Another works incorporated derandomized JL Transforms (Acharya et al., 2019). However, setups in the highlighted works did not capitalize preserving distances between vectors, which did not favour the usage of JL Transforms over the sparsifications or sketchings.

In this manuscript, we connect distributed optimization with the $l_2$ property of JL Transforms, that guarantees distances between vectors will not change drastically. Another important side effect of this approach is the derived convergence with high probability. It assures, that the desired criterion is satisfied with high confidence, rather than averagely.

**Our contribution.** In this work, we

• Compare JL mappings with sparsification compressors, such as random coordinates choice.

• Explore the application of JL Transforms to the aforementioned distributed and federated learning scenarios, by modifying the existing algorithms.

• Demonstrate their effectiveness in reducing communication costs without compromising the convergence on well-established benchmark datasets.

• Provide theoretical convergence rates for the adjusted methods.

• Validate our results with experiments.

## 2 MAIN PART

### 2.1 NOTATION AND PRELIMINARIES

We use the standard Euclidean norm for vectors: $\|x\| \stackrel{\text{def}}{=} \langle x, x \rangle^{1/2}, x \in \mathbb{R}^n$. The objective functional $f : \mathbb{R}^n \to \mathbb{R}$ is a differentiable function. Below we introduce all the definitions that are used throughout the manuscript.

**Definition 1** (Smoothness). *Differentiable function $f$ has L-Lipschitz gradient, i.e.*
$$\|\nabla f(x) - \nabla f(y)\| \leq L\|x - y\| \quad \forall x, y \in \mathbb{R}^n.$$

**Definition 2** (Convexity). *Differentiable function $f$ is convex, i.e.*
$$f(y) \geq f(x) + \langle \nabla f(x), y - x \rangle, \quad \forall x, y \in \mathbb{R}^n.$$

**Definition 3** (Strong convexity). *Differentiable function $f$ is $\mu$-strongly-convex, i.e.*
$$f(y) \geq f(x) + \langle \nabla f(x), y - x \rangle + \frac{\mu}{2}\|y - x\|^2, \quad \forall x, y \in \mathbb{R}^n.$$

### 2.2 JL TRANSFORMS

**Definition 4** (JL Transform (Johnson et al., 1984)). *For every $\varepsilon, \delta \in (0, 1)$ there exists a stochastic mapping $h : \mathbb{R}^n \to \mathbb{R}^k$ is a JL Transform, i.e. $\forall u, v \in \mathbb{R}^n$*
$$\mathbb{P}\left[(1 - \varepsilon)\|u - v\|^2 \leq \|h(u) - h(v)\|^2 \leq (1 + \varepsilon)\|u - v\|^2\right] \geq 1 - \delta$$

It turns out, that if the mapping is linear (e.g. multiplying by a matrix), JL Transform also preserve the scalar products

**Lemma 1.** *If $h$ is a linear JL Transform, then,*
$$\mathbb{P}\left[\,|\langle h(u), h(v)\rangle - \langle u, v\rangle| \leq \varepsilon\|u\|\,\|v\|\,\right] \geq 1 - 2\delta, \quad u, v \in \mathbb{R}^n.$$

We analyze two linear types of JL Transforms – continuous and discrete ones.

**Definition 5** (Gaussian JL matrix). *Let $S \in \mathbb{R}^{k \times n}$ and for all $i \in [k]$, $j \in [n]$ we have*
$$[S]_{ij} \sim \mathcal{N}\left(0, \frac{1}{k}\right)$$
*and i.i.d.. Given, $\varepsilon, \beta > 0$, if $k \geq \frac{4 + 2\beta}{\varepsilon^2/2 - \varepsilon^3/3} \log m$, then $S$ satisfy Definition 4 with $\varepsilon, \delta = \frac{1}{m^\beta}$ (Dasgupta & Gupta, 2003).*

**Definition 6** (Rademacher JL matrix). *Let $S \in \mathbb{R}^{k \times n}$ and for all $i \in [k]$, $j \in [n]$ we have*
$$[S]_{ij} = \begin{cases} \frac{1}{\sqrt{k}}, & p = 1/2, \\ -\frac{1}{\sqrt{k}}, & p = 1/2, \end{cases}$$
*and i.i.d.. Given, $\varepsilon, \beta > 0$, if $k \geq \frac{4 + 2\beta}{\varepsilon^2/2 - \varepsilon^3/3} \log m$, then $S$ satisfy Definition 4 with $\varepsilon, \delta = \frac{1}{m^\beta}$ (Achlioptas, 2003).*

**Corollary 1.** *For arbitrary $\varepsilon, \delta$ Gaussian and Rademacher JL matrix, we need $k = \Theta\left(\frac{\log 1/\delta}{\varepsilon^2}\right)$*

However, since we decrease the communications costs via the dimension reduction, one should be able to decompress the vectors to iterate. We obstacle a mismatch between dimensions – $x \in \mathbb{R}^n$, whereas $Sx \in \mathbb{R}^k$. To overcome this issue we have to invert the mapping efficiently. Therefore, we introduce the operator
$$JL_S(x) \stackrel{\text{def}}{=} S^T Sx.$$
It is obvious, that $JL_S(x) \in \mathbb{R}^n$. It turns out, that for matrices introduced above, this operator satisfy several condition.

**Lemma 2** (Unbiasedness). *For stochastic matrices $S$ as in Definitions 5 and 6 the operator $JL_S$ is unbiased, i.e.*
$$\mathbb{E}JL_S(x) = x, \quad \forall x \in \mathbb{R}^n.$$

**Lemma 3** (Variance). *For Gaussian JL matrix $S_{Gauss}$ and Rademacher $S_{Rad}$, operator $JL_S$ satisfy following property:*
$$\mathbb{E}\|JL_{S_{Gauss}}(x)\|_2^2 = \left(\frac{n + k + 1}{k}\right)\|x\|_2^2,$$

$$\mathbb{E}\,\|JL_{S_{Rad}}(x)\|_2^2 = \left(\frac{n+k-1}{k}\right)\|x\|_2^2, \quad \forall x \in \mathbb{R}^n.$$

Proofs can be seen in Appendix.

In terms of Lemmas above, it can be seen, that operator $JL_S$ is similar to unbiased sparsification compressor to some degree. Consider, for instance, `Randk`:

$$\mathcal{C}(x) = \frac{n}{k}\sum_{i \in S}x_i e_i,$$

where $k \in [n]$ and $S \subset [n]$ is the $k$-nice sampling; i.e. a subset of $[n]$ of cardinality $k$, chosen uniformly at random. Vectors $e_1, \ldots, e_n$ are the standard unit basis vectors in $\mathbb{R}^n$. From the definition it can be noticed, that `Randk` sends $k$ coordinates, instead of $n$, moreover, it is unbiased with variance $\frac{n}{k}$ (Beznosikov et al., 2023), as $JL_S$.

However, it can be shown, that these sparsifications do not preserve vector's norm with high probability:

**Example 1.** *Let $\mathcal{C}$ be a* `Randk` *compressor. Then, for any $x \in \mathbb{R}^n$*

$$\|\mathcal{C}(x)\|^2 = \frac{n^2}{k^2}\sum_{i \in S}x_i^2.$$

*We aim to analyze Definition 4:*

$$\mathbb{P}\left[(1-\varepsilon)\|u-v\|^2 \le \|\mathcal{C}(u)-\mathcal{C}(v)\|^2 \le (1+\varepsilon)\|u-v\|^2\right].$$

*For the sake of simplicity we assume the compressor is applied to vectors $u$ and $v$ with the same seed. Therefore, we use linearity and rewrite this expression in the following way:*

$$\mathbb{P}\left[(1-\varepsilon)\|u-v\|^2 \le \|\mathcal{C}(u-v)\|^2 \le (1+\varepsilon)\|u-v\|^2\right],$$

*or*

$$\mathbb{P}\left[\left|\frac{\|\mathcal{C}(u-v)\|^2}{\|u-v\|^2} - 1\right| \le \varepsilon\right].$$

*As $u,v \in \mathbb{R}^n$ are arbitrary, we end up with*

$$\mathbb{P}\left[\left|\frac{\|\mathcal{C}(x)\|^2}{\|x\|^2} - 1\right| \le \varepsilon\right], \quad \forall x \in \mathbb{R}^n. \tag{8}$$

*Below we consider various $x$ and try to estimate the $\varepsilon$ and $\delta$ for the* `Randk` *compressor.*

*1) If $x = (1,1,\ldots,1) \in \mathbb{R}^n$, then,*

$$\|x\|^2 = n, \quad \|\mathcal{C}(x)\|^2 = \frac{n^2}{k}.$$

*As $\varepsilon \in (0,1)$ is always less than $\frac{n}{k}$, the probability (8) is 0. And we are interested in taking small $\varepsilon$, since they mean smaller deviations in norm from considered vectors.*

*2) If $x = (1,0,\ldots,0) \in \mathbb{R}^n$, then*

$$\|x\|^2 = 1, \quad \|\mathcal{C}(x)\|^2 = \begin{cases} \frac{n^2}{k^2}, & p = \frac{k}{n}, \\ 0, & p = 1 - \frac{k}{n}. \end{cases}$$

*Since $\varepsilon \in (0,1)$ is always less than $\frac{n^2}{k^2}$, the probability (8) is $1 - \frac{k}{n}$.*

*Overall, these examples demonstrate, that random sparsification cannot be described with $\varepsilon$ and $\delta$ for any vector $x$, as it is required in Definition 4.*

The next lemma will be helpful in the high probability convergence analysis.

**Lemma 4.** *For matrices $S$, as in Definition 4, the operator $JL_S$ satisfy following property:*

$$\|JL_S(x)\|^2 \le (1+\varepsilon)^2\|x\|^2$$

*with probability $1-\delta$.*

As shown above, random sparsifications do not have this feature, therefore JL Transforms are more preferable in high probability analysis.

Further we adjust the existing methods to the proposed mappings. We may use the same matrix $S$ throughout the algorithm, or generate the new one at every iteration. In the latter option the same

matrix is needed to be maintained at every computing device. This can be done by assigning the same random variable generator to devices with the same initial seeds.

## 2.3 BYZANTINE ATTACKS

Various algorithms incorporate the gradient similarity in Byzantine-resilient optimization. We focus on the algorithms `Grad-BANT`, inspired by (Yan et al., 2023). Method, described in (2), utilize the idea of the trial function $f_0(x)$, which is stored on the server, which is an honest device. It is evaluated on a data distribution, that matches the overall homogeneous distribution. It is not vulnerable to adversarial attacks, hence can be trusted. Therefore, we compare our local gradients to the gradient of the trial function, in order to decide to define the client as a Byzantine or not.

For more details one may look at Algorithm 1, where all the required steps are described.

This method heavily rely on the core idea of assigning trust scores to devices. At every iteration we aggregate the devices' gradients $g_i^t$ (line 5), aiming to aggregate them and conduct a step (line 12). However, since some of the clients may send corrupted gradients, we are interested in adjusting the contribution coefficients for each worker at each step. If $\langle \nabla f_0(x^t), g_i^t \rangle > \frac{L\gamma}{2}\|g_i^t\|^2$, the gradient does not misdirect the overall gradient direction, hence it should be assigned some positive weight. Otherwise, it should not be assigned any weight at all. We ensure non-negativity with $[x]_0 \overset{\text{def}}{=} \max\{x, 0\}$ and normalize, in order to maintain the overall sum of weights. If all gradients are not co-directed with the main direction, we assign them equal weights, in order to preserve convergence (line 10). Adding momentum $\beta$ makes our scheme more

---

**Algorithm 1** `Grad-BANT`

1: **Parameters:** $x^0 \in \mathbb{R}^n$, $\omega_i^0 = 1/n$, stepsize $\gamma$, momentum $\beta \in [0, 1]$, smoothness constant $L$
2: **for** $t = 0, 1, \ldots, T-1$ **do**
3:     Generate $S_t$; server sends $x^t$ to workers
4:     **for** all $i = 1, \ldots, n$ in parallel **do**
5:         $g_i^t = \nabla f_i(x^t)$; send $S_t g_i^t$ to server
6:     **end for**
7:     $\theta_i^t = \langle S_t \nabla f_0(x_t), S_t g_i^t \rangle - \frac{L\gamma}{2}\|S_t^T S_t g_i^t\|^2$
8:     $\omega_i^t = (1-\beta)\omega_i^{t-1} + \beta \frac{[\theta_i^t]_0}{\sum_{j=1}^n [\theta_j^t]_0}$
9:     **if** all $[\theta_i^t]_0 = 0$ **then**
10:        $\omega_i^t = (1-\beta)\omega_i^{t-1} + \frac{\beta}{n}$
11:     **end if**
12:     $x^{t+1} = x^t - \gamma S_t^T \sum_{i=1}^n \mathbb{I}_{[\theta_i^t > 0]} \omega_i^t S_t g_i^t$
13: **end for**

---

stable, as it allows previous good gradients to influence the current step (lines 8 and 10). Also, it is applicable to strategies, where Byzantine agents mix poison messages with honest one, as we maintain the impact of decent gradients and not include the corrupted ones.

In order to transfer less bits, we utilize the JL Transforms in the communication scheme. Therefore, in order to match the dimensions we investigate $\langle S_t \nabla f_0(x^t), S_t g_i^t \rangle$ (line 8). With high probability it does not deviate too much from the exact scalar product, hence, does not disrupt the convergence. In order to return to the initial dimension we invert mapping by multiplying it with $S_t^T$ (line 12). This approach prevents Byzantine agents from compromising the training process by transmitting less bits.

In this setting, we need following property from local devices:

**Assumption 1** (Data similarity). *For distributed optimization local worker possess* $(\delta_1, \delta_2)$-*data similarity, i.e.*
$$\|\nabla f_i(x) - \nabla f(x)\|^2 \leq \delta_1 + \delta_2 \|\nabla f(x)\|^2, \quad \forall x \in \mathbb{R}^n.$$

The theoretical convergence bounds are the following:

**Theorem 1.** *Let $f_0$ be $L$-smooth, convex, local datasets satisfy Assumption 1 with $\delta_2 \leq \frac{1}{16}$, $x_* = \arg\min_x f_0(x)$ and JL Transforms are applied. Then, with $\gamma \leq \frac{1}{16L}$ after $T$ iterations the following holds:*
$$\min_{1 \leq t \leq T} \|\nabla f(x^t)\|^2 = \mathcal{O}\left( \frac{(f_0(x^0) - f_0(x_*)) L}{T} + \delta_1 \right),$$
*with probability $1 - \delta$, where $k \geq \frac{4 \log \frac{(2M+1)T}{\delta}}{\varepsilon^2/2 - \varepsilon^3/3}$.*

This correlates with the result without compression, achieved in (Molodtsov et al., 2025), where they claim $\min_{1 \leq t \leq T} \mathbb{E}\|\nabla f(x^t)\|^2 = \mathcal{O}\left( \frac{(f_0(x^0) - f_0(x_*)) L}{T} + \delta_1 \right)$.

Typically, employing an unbiased compressor operator with variance $\omega \geq 1$ introduces an additional multiplicative factor of $\omega$ (Mishchenko et al., 2019; Gorbunov et al., 2021; Tyurin & Richtárik, 2022). For example, combining gradient descent with random coordinate selection increases the overall complexity by a factor of $\frac{d}{k}$, since for the Rand-$k$ compressor we have $\omega = \frac{d}{k}$. In contrast, our analysis not only avoids this overhead but also provides high-probability guarantees, which are particularly valuable as they reduce uncertainty.

## 2.4 Personalization

In the personalization setting, we are dealing with (3). We stick to the formalization in (Hanzely et al., 2020), where the authors provide optimal algorithms in terms of the communications.

The idea is to regularize the deviation between the local model and the global one, that accumulates and averages the weights. This approach is similar to the `FedProx` algorithm (Li et al., 2020). Every client conduct local steps in order to obtain a reasonable solution for an intermediate problem (line 7). After that, workers exchange their local states to compute the average and to proceed to the next iteration (lines 4 and 10). This allows, to find optimum iteratively, taking into account the average model's state. One of the benefits of this algorithm is the suboptimal solution to the local problem - we do not need the exact minimizer, as it might not be achievable in observable time. However, we can conduct limited number of local subsolver and still have a decent convergence.

We utilize the JL Transforms in transmitting the models' states, since we are interested in the $l_2$-difference between the aggregated condition and the local one. Therefore, we transmit significantly less bits and achieve following theoretical bounds:

**Theorem 2.** *Let functions $f_i$ be $L$-smooth, $\mu$-strongly convex and $f_i \geq 0$ for all $i$. Let AGD with starting point $y_i^t$ be employed for $J_t \overset{\text{def}}{=} \sqrt{\frac{L+\lambda}{\mu+\lambda}} \log\left( 1152 L\lambda M^2 \left( 2\frac{\lambda}{\mu} + 1 \right)^2 \mu^{-2} \right) +$*

$4\sqrt{\frac{\mu(L+\lambda)}{\lambda(\mu+\lambda)}} t$ *iterations to approximately solve subproblem at iteration $t$ and $T$ iterations overall. Define $f_* = \arg\min_x f(x)$ Then, we have*

$$f(x^T) - f_* = \mathcal{O}\left( \left( 1 - \sqrt{\frac{\mu}{\lambda}} \right)^T \left( f(x^0) - f_* \right) \right)$$

*with probability $1 - \delta$, where $k \geq \frac{4 \log \frac{2MT}{\delta}}{\varepsilon^2/2 - \varepsilon^3/3}$.*

This correlates with the result without compression, obtained in (Hanzely et al., 2020), where they achieve $f(x^T) - f_* = \mathcal{O}\left( \left( 1 - \sqrt{\frac{\mu}{\lambda}} \right)^T \left( f(x^0) - f_* \right) \right)$. Moreover, in our results number of needed local iterations $\left( \sum_{t=1}^T J_t \right)$ does not differ from one, established in (Hanzely et al., 2020).

## 2.5 Vertical FL

The problem (4) is solved via the `ADMM` algorithm, which is based on augmented Lagrangian method. We inspect the scaled version (Section 3.1.1 from (Boyd et al., 2011)) due to the ease of notation.

---

**Algorithm 2** APGD

1: **Parameters:** $y^0 \in \mathbb{R}^{kd}$, $x^0 \in \mathbb{R}^{nd}$, regularization $\lambda$, strong convexity $\mu$
2: **for** $t = 0, 1, \ldots, T-1$ **do**
3:     Central server computes $\overline{y}^t = \frac{1}{M} \sum_{i=1}^M y_i^t$
4:     Send $\overline{y}^t$ to clients
5:     **for** all nodes $i = 1, \ldots, M$ in parallel **do**
6:         Set $h_i^{t+1}(z) = f_i(z) + \frac{\lambda}{2}\|S_t z - \overline{y}^t\|^2$
7:         Using AGD find $x_i^{t+1}$ s.t.
$$h_i^{t+1}(x_i^{t+1}) \leq \varepsilon_t + \min_z h_i^{t+1}(z)$$
8:         Generate $S_{t+1}$
9:         Update
$$y_i^{t+1} = S_{t+1}\left( x_i^{t+1} + \frac{\sqrt{\lambda} - \sqrt{\mu}}{\sqrt{\lambda} + \sqrt{\mu}} \left( x_i^{t+1} - x_i^t \right) \right)$$
10:         Send $y_i^{t+1}$ to server
11:     **end for**
12: **end for**

---

**Algorithm 3** AGD

1: **Parameters:** $x^0 = y^0 \in \mathbb{R}^n$, step size $\gamma$, momentum $\beta \in [0, 1]$
2: **for** $j = 0, 1, \ldots, J$ **do**
3:     $x^{j+1} = x^j - \gamma \nabla f(y^j)$
4:     $y^{j+1} = x^{j+1} + \beta \left( x^{j+1} - x^j \right)$
5: **end for**

---

---

**Algorithm 4** D-ADMM

---

1: **Parameters:** Starting points $x_i^0 \in \mathbb{R}^{n_i}$, $\forall i \in [M]$, $\overline{z}^0 \in \mathbb{R}^m$, $u^0 \in \mathbb{R}^k$, JL matrix $S \in \mathbb{R}^{k \times m}$.
2: **for** $t = 0, 1, 2, \ldots, T - 1$ **do**
3:      Solve $\overline{z}^{t+1} = \arg\min\limits_{z \in \mathbb{R}^m} \left\{ l(Mz; b) + (M\rho/2) \left\| Sz - \widehat{Ax}^t - u^t \right\|_2^2 \right\}$
4:      Update $u^{t+1} = u^t + \widehat{Ax}^t - S\overline{z}^{t+1}$
5:      Send $u^{t+1}, \widehat{Ax}^t, \hat{z}^{t+1} = S\overline{z}^{t+1}$ to nodes
6:      **for** all nodes $i = 1, 2, \ldots, M$ in parallel **do**
7:          Solve $x_i^{t+1} = \arg\min\limits_{x \in \mathbb{R}^{n_i}} \left\{ r_i(x) + (\rho/2) \left\| SA_i(x - x_i^t) - \hat{z}^{t+1} + \widehat{Ax}^t + u^{t+1} \right\|_2^2 \right\}$
8:          Send $\widehat{A_i x_i}^{t+1} = SA_i x_i^t$ to server
9:      **end for**
10:     Aggregate $\widehat{Ax}^{t+1} = \frac{1}{M} \sum\limits_{i=1}^M \widehat{A_i x_i^t}$
11: **end for**

---

Therefore, due to the choice of variables, each node can independently solve a minimization problem (line 7), and exchange their solutions with the central server. After aggregating we are able to iterate another descent (line 3) and update the dual variable (line 4). To mitigate the communication costs we use the JL mappings (lines 5, 8) , hence, the subproblems' solutions do not differ much.

**Theorem 3.** *Let functions $l$ and $r$ be convex. Define* $\overline{x}^k = \frac{1}{M} \sum_{i=1}^M x_i^k$, $\widetilde{x}^T = \frac{1}{T+1} \sum_{k=0}^T \overline{x}^k$, $\widetilde{z}^T = \frac{1}{T+1} \sum_{k=0}^T \overline{z}^k$, $\widetilde{u}^T = \frac{1}{T+1} \sum_{k=0}^T u^k$, $F(x, z, u) = (-A^T u, -u, Ax - z)$, $(x_*, z_*)$ *- optimal point and* $u_*$ *- dual optimal point. Then, after $T$ iterations Algorithm 4 achieves*

$$\left( l(\widetilde{x}^T) + r(\widetilde{z}^T) \right) - \left( l(x_*) + r(z_*) \right)$$

$$+ \left\langle \left( \widetilde{x}^T - x_*, \ \widetilde{z}^T - z_*, \ \widetilde{u}^T - u_* \right), \ F(x_*, z_*, u_*) \right\rangle$$

$$= \mathcal{O}\left( \frac{\|\overline{x}^0 - x_*\|^2 \ + \ \|\overline{z}^0 - z_*\|^2 \ + \ \|u^0 - u_*\|^2}{T} \right),$$

*with probability $1 - \delta$.*

These bounds coincide with the results from (He & Yuan, 2012), where no dimension reduction is applied. This is achieved, as any JL Transform preserve the feasibility region with high probability.

## 3 EXPERIMENTS

In this section, we provide experimental results that validate the theoretical contributions of our work. Specifically, we compare two variants of stochastic projection matrices: Gaussian and Rademacher one.

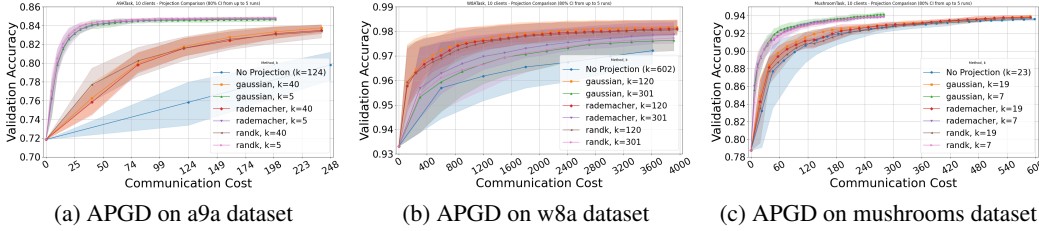

| (a) APGD on a9a dataset | (b) APGD on w8a dataset | (c) APGD on mushrooms dataset |

Figure 1: APGD performance with JL Transforms

**Byzantine-robust Optimization.** Further we investigate the effect of the random projections in the Byzantine setting on ResNet-20 training on CIFAR-10. We compare the Rademacher (6) and Gaussian (5) JL matrices with Randk compression. At Figure **??** % stands for the fraction

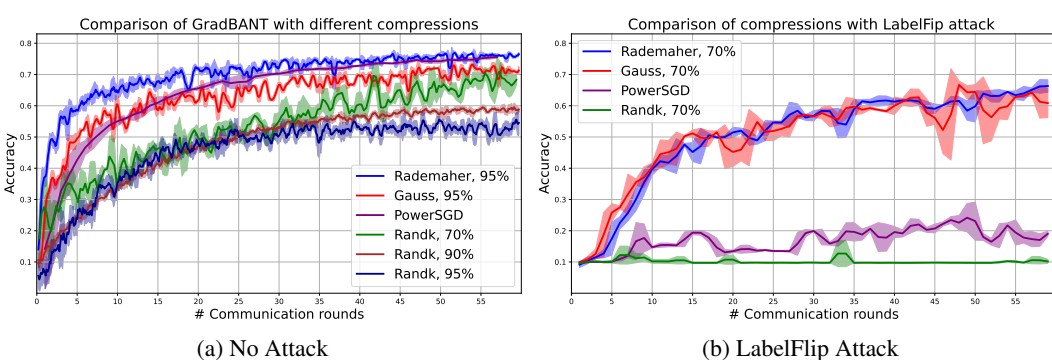

(a) No Attack

(b) LabelFlip Attack

Figure 2: Grad-BANT performance with JL Transforms

of **compressed** coordinates. Hence, our proposed methods not only outscores random coordinate sketching, but also consumes 5 times less bandwidth. More experiments, including evaluating the robustness against byzantine attacks can be found in Appendix.

**Personalized Federated Learning.** We evaluate the performance of personalized federated learning using `APGD` with compressions, provided by JL projections. We demonstrate that dimensionality reduction via JL Transforms improves convergence efficiency, enabling higher model accuracy with much fewer transmitted bits. The number of clients $M = 10$.

**Vertical Federated Learning.** Last experiments demonstrate the performance of JL Transforms in vertical FL, using a distributed `ADMM` method. We conduct experiments on the linear regression problem: $\min_{x \in \mathbb{R}^n} \left[ f(x) = \frac{1}{2}\|Ax - b\|^2 + \lambda\|x\|^2 \right]$. We take `mushrooms` and `phishing` dataset from LibSVM (Chang & Lin, 2011) library and $\lambda = 1$. We also compare the performance with `RandK` operator and the scenario without any compressed messages.

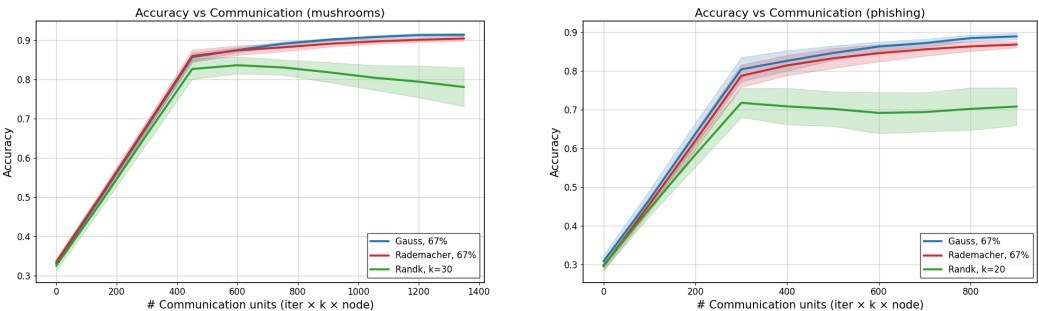

Figure 3: D-ADMM performance with JL Transforms.

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

## A    TECHNICAL STATEMENTS

**Lemma 5** (Preserve of scalar product). *If $h$ is a linear JL Transform, then, with probability $1 - 2\delta$ we have*

$$|\langle h(x), h(y)\rangle - \langle x, y\rangle| \le \varepsilon\|x\| \cdot \|y\|.$$

*Proof.* If at least one of $x$ and $y$ is the 0-vector, then this is trivially satisfied. If $x$ and $y$ are both unit vectors then we assume w.l.o.g. that $\|x + y\|^2 \ge \|x - y\|^2$ and we proceed as follows:

$$
\begin{aligned}
4|\langle h(x), h(y)\rangle - \langle x, y\rangle| &= |\|h(x) + h(y)\|^2 - \|h(x) - h(y) - 4\langle x, y\rangle| \\
&\le |(1 + \varepsilon)\|x + y\|^2 - (1 - \varepsilon)\|x - y\|^2 - 4\langle x, y\rangle| \\
&= |4\langle x, y\rangle + \varepsilon(\|x + y\|^2 + \|x - y\|^2) - 4\langle x, y\rangle| \\
&= \varepsilon(2\|x\|^2 + 2\|y\|^2) \\
&= 4\varepsilon.
\end{aligned}
$$

Otherwise we can reduce to the unit vectors:

$$
\begin{aligned}
\langle h(x), h(y)\rangle - \langle x, y\rangle| &= \left|\left\langle h\left(\frac{x}{\|x\|}\right), h\left(\frac{y}{\|y\|}\right)\right\rangle - \left\langle \frac{x}{\|x\|}, \frac{y}{\|y\|}\right\rangle\right| \|x\| \cdot \|y\| \\
&\le \varepsilon\|x\| \cdot \|y\|.
\end{aligned}
$$

As we twice use the property of the JL matrix, we have $1 - 2\delta$ probability. $\square$

**Lemma 6.** *Let $S \in \mathbb{R}^{k \times n}$ be a Gaussian JL matrix. Then,*

$$\mathbb{E}S^T S x = x,$$

*and*

$$\mathbb{E}\|S^T S x\|^2 = \left(\frac{n + k + 1}{k}\right)\|x\|^2.$$

*Proof.* We will start with the unbiasedness. If $[S]_{i,j} \sim \mathcal{N}(0, \frac{1}{k})$, then,

$$[S^T S]_{i,j} = \sum_{r=1}^{k}[S]_{i,r}[S]_{j,r}$$

If $i \ne j$, then, from independence we have

$$
\mathbb{E}[S^T S]_{i,j} = \begin{cases} \mathbb{E}\sum_{r=1}^{k}\mathcal{N}(0, \frac{1}{k})^2, & i = j \\ 0, & i \ne j. \end{cases}
$$

We use the fact, that $\mathcal{N}(0, \frac{1}{k}) = \frac{1}{\sqrt{k}}\mathcal{N}(0, 1)$, and that $\mathbb{E}\mathcal{N}(0, 1)^2 = 1$. Therefore,

$$
\mathbb{E}[S^T S]_{i,j} = \begin{cases} 1, & i = j \\ 0, & i \ne j, \end{cases}
$$

and $\mathbb{E}S^T S x = x$.

For the variance bound we might write following:

$$[S^T S x]_i = \sum_{j=1}^{n}\sum_{r=1}^{k}[S]_{i,j}[S]_{j,r}x_j,$$

and

$$[S^T S x]_i^2 = \sum_{j,l}^{n}\sum_{r,m}^{k}[S]_{i,r}[S]_{j,r}[S]_{i,m}[S]_{l,m}x_j x_l.$$

After taking the expectation, we will maintain terms with 1) $i = j = l$ and $r = m$, 2)$i = j = l$ and $r \ne m$, 3)$j = l, r = m$ and $i \ne j$. Then,

1. $\mathbb{E}[S^T S x]_i^2 = \sum_{r=1}^{k}\mathbb{E}\mathcal{N}(0, \frac{1}{k})^4 x_i^2 = \sum_{r=1}^{k}\frac{3}{k^2}x_i^2 = \frac{3}{k}x_i^2,$

2. $\mathbb{E}[S^T S x]_i^2 = \sum\limits_{r=1}^{k} \sum\limits_{m \neq r}^{k} \frac{1}{k} \frac{1}{k} x_i^2 = \frac{k(k-1)}{k^2} x_i^2 = \frac{k-1}{k} x_i^2,$

3. $\mathbb{E}[S^T S x]_i^2 = \sum\limits_{j \neq i}^{n} \sum\limits_{r=1}^{k} \frac{1}{k} \frac{1}{k} x_j^2 = \sum\limits_{j \neq i}^{n} \frac{1}{k} x_j^2 = \frac{1}{k} \|x\|^2 - \frac{1}{k} x_i^2.$

Summing over $i$ we achieve

$$\mathbb{E}\|S^T S x\|^2 = \left( \frac{3}{k} + \frac{k-1}{k} + \frac{n-1}{k} \right) \|x\|^2 = \left( \frac{n+k+1}{k} \right) \|x\|^2.$$

$\square$

**Lemma 7.** *Let $S \in \mathbb{R}^{k \times n}$ be a Rademacher JL matrix. Then,*
$$\mathbb{E} S^T S x = x,$$
*and*
$$\mathbb{E}\|S^T S x\|^2 = \left( \frac{n+k+1}{k} \right) \|x\|^2.$$

*Proof.* For the ease of notation we will introduce the notation
$$Rad\,(1/2) = \begin{cases} 1, & p = 1/2, \\ -1, & p = 1/2. \end{cases}$$
We will start with the unbiasedness. If $[S]_{i,j} \sim \frac{1}{\sqrt{k}} Rad(1/2)$, then,
$$[S^T S]_{i,j} = \sum_{r=1}^{k} [S]_{i,r} [S]_{j,r}$$
If $i \neq j$, then, from independence we have
$$\mathbb{E}[S^T S]_{i,j} = \begin{cases} \mathbb{E} \sum\limits_{r=1}^{k} \frac{1}{k} Rad(1/2)^2, & i = j \\ 0, & i \neq j. \end{cases}$$
We use the fact, that $Rad(1/2)^2$ takes values only equals to 1. Therefore,
$$\mathbb{E}[S^T S]_{i,j} = \begin{cases} 1, & i = j \\ 0, & i \neq j, \end{cases}$$
and $\mathbb{E} S^T S x = x.$
For the variance bound we might write following:
$$[S^T S x]_i = \sum_{j=1}^{n} \sum_{r=1}^{k} [S]_{i,j} [S]_{j,r} x_j,$$
and
$$[S^T S x]_i^2 = \sum_{j,l}^{n} \sum_{r,m}^{k} [S]_{i,r} [S]_{j,r} [S]_{i,m} [S]_{l,m} x_j x_l.$$
After taking the expectation, we will maintain terms with 1) $i = j = l$ and $r = m$, 2)$i = j = l$ and $r \neq m$, 3)$j = l, r = m$ and $i \neq j$. Then,

1. $\mathbb{E}[S^T S x]_i^2 = \sum\limits_{r=1}^{k} \frac{1}{k^2} \mathbb{E} Rad(1/2)^4 x_i^2 = \sum\limits_{r=1}^{k} \frac{1}{k^2} x_i^2 = \frac{1}{k} x_i^2,$

2. $\mathbb{E}[S^T S x]_i^2 = \sum\limits_{r=1}^{k} \sum\limits_{m \neq r}^{k} \frac{1}{k} \frac{1}{k} x_i^2 = \frac{k(k-1)}{k^2} x_i^2 = \frac{k-1}{k} x_i^2,$

3. $\mathbb{E}[S^T S x]_i^2 = \sum\limits_{j \neq i}^{n} \sum\limits_{r=1}^{k} \frac{1}{k} \frac{1}{k} x_j^2 = \sum\limits_{j \neq i}^{n} \frac{1}{k} x_j^2 = \frac{1}{k} \|x\|^2 - \frac{1}{k} x_i^2.$

Summing over $i$ we achieve

$$\mathbb{E}\|S^T S x\|^2 = \left(\frac{1}{k} + \frac{k-1}{k} + \frac{n-1}{k}\right)\|x\|^2 = \left(\frac{n+k-1}{k}\right)\|x\|^2.$$

$\square$

Statement below is crucial for deriving convergence for proposed methods, since:

**Lemma 8.** *Suppose we have events $E_1, E_2, \ldots, E_n$ that holds with following probabilities:*
$$\mathbb{P}\left[E_i\right] \geq 1 - \delta_i,$$
*Then,*
$$\mathbb{P}\left[\bigcap_{i=1}^{n} E_i\right] \geq 1 - \sum_{i=1}^{n} \delta_i$$

*Proof.*

$$\mathbb{P}\left[\bigcap_{i=1}^{n} E_i\right] = 1 - \mathbb{P}\left[\bigcup_{i=1}^{n} \overline{E_i}\right] \geq 1 - \sum_{i=1}^{n} \mathbb{P}\left[\overline{E_i}\right] \geq 1 - \sum_{i=1}^{n} \delta_i$$

$\square$

Next lemma will be used in Byzantine setting:

**Lemma 9.** *Let $S$ be a JL Matrix. Then,*
$$\left\|S^T S x\right\|^2 \leq (1 + \varepsilon)^2 \|x\|^2$$
*with probability at least $1 - \delta$.*

*Proof.* From JL property with probability $1 - \delta$ we have
$$(1 - \varepsilon)\|x\|^2 \leq \|Sx\|^2 \leq (1 + \varepsilon)\|x\|^2,$$
therefore, eigenvalues of $S^T S$ are located in $[1 - \varepsilon, 1 + \varepsilon]$. On the other hand, we have
$$\|S^T S x\| \leq \sigma_{max}(S^T S)\|x\|.$$
Since $S^T S$ is symmetric, its singular values are the absolute values of its eigenvalues, therefore, $\sigma_{max}(S^T S) \leq 1 + \varepsilon$. And
$$\|S^T S x\|^2 \leq (1 + \varepsilon)^2 \|x\|^2$$

$\square$

Next lemma will be useful throughout some theorems, allowing us to deal with norms:

**Lemma 10** (Young's inequality)**.** *Let $x, y \in \mathbb{R}^n$. Then, for any $\alpha > 0$ we have*
$$\langle x, y \rangle \leq \frac{1}{2\alpha}\|x\|^2 + \frac{\alpha}{2}\|y\|^2.$$

## B BYZANTINE ATTACKS

We follow the analysis given in (Molodtsov et al., 2025), adjusting it to our needs. With the trial function $f_0$ we introduce the server function $f_1$. Note, that server is an honest device. Also, we introduce the number $\zeta(N)$, which reflects the relationship between $f_1$ and $f_0$, where $N$ is the size of trial dataset.

**Lemma 11.** *Suppose $f_0$ is $L$-smooth. Then for all $x \in \mathcal{X} \subset \mathbb{R}^d$ with probability of at least $1 - \widetilde{\delta}$ over a sample of size $N$, the following estimate, linking the trial function with the objective function on the server, is valid:*

$$\|\nabla f_1(x) - \nabla f_0(x)\|_2^2 \leqslant \zeta(N) = \widetilde{\mathcal{O}}\left(\frac{1}{N}\right),$$

*Proof.* Given the norm inequality $\| \cdot \|_2 \leqslant \sqrt{d} \cdot \| \cdot \|_\infty$, we can recast the scalar product in the following manner:

$$\|\nabla f_1(x) - \nabla f_0(x)\|_2^2 \quad \leqslant \quad d \cdot \|\nabla f_1(x^t) - \nabla f_0(x)\|_\infty^2. \tag{9}$$

To establish the uniform convergence of $\|\nabla f_1(x) - \nabla f_0(x)\|_\infty^2$, we employ Theorem 5 from (**?**). This theorem provides a bound on the $\ell_\infty$-covering number of the function class $\mathcal{F} = \{\xi \mapsto \nabla f_1(x;\xi) \mid x \in \mathcal{X}\}$. Given that $\mathcal{X}$ resides within an $\ell_2$-sphere, let us define it bound by $S$, the covering number for $\mathcal{X}$ using the Euclidean metric $d_2(x_i, x_j) = \|x_i - x_j\|_2$ is constrained as follows for $d > 3$:

$$N(\varepsilon, \mathcal{X}, d_2) = \mathcal{O}\left(d^2\left(\frac{S}{\varepsilon}\right)^d\right).$$

In evaluating the covering numbers for $\mathcal{F}$ under the $\ell_\infty$ metric, where $\|\nabla f_1(x_i;\cdot) - \nabla f_1(x_j;\cdot)\|_\infty = \sup_\xi |\nabla f_1(x_i;\xi) - \nabla f_1(x_j;\xi)|$, the $L$-smoothness property facilitates the following assertion:

$$\forall x_i, x_j \in \mathcal{X} \hookrightarrow \|\nabla f_1(x_i;\cdot) - \nabla f_1(x_j;\cdot)\|_\infty \leqslant \|\nabla f_1(x_i;\cdot) - \nabla f_1(x_j;\cdot)\|_2 \leqslant L\|x_i - x_j\|.$$

This indicates that an $\varepsilon$-net for $\mathcal{X}$ in $d_2$ space concurrently serves as an $L\varepsilon$-net for $\mathcal{F}$ in $d_\infty$ space:

$$N(\varepsilon, \mathcal{F}, d_\infty) \leqslant N(\varepsilon/L, \mathcal{X}, d_2) = \mathcal{O}\left(d^2\left(\frac{LS}{\varepsilon}\right)^d\right).$$

Following this analysis, we derive an estimation consistent with the findings in (**?**):

$$\|\nabla f_1(x) - \nabla f_0(x)\|_\infty^2 = \widetilde{\mathcal{O}}\left(\frac{1}{N}\right).$$

Defining the notation

$$\zeta(N) \stackrel{\text{def}}{=} \widetilde{\mathcal{O}}\left(\frac{1}{N}\right),$$

and substituting this into equation 9 concludes the proof of the lemma.

$\square$

This lemma is technical in nature, and we significantly benefit from the assertion established in the previous lemma. Ultimately, we derive an important estimate for the scalar product, which appears in many subsequent proofs throughout this work.

**Lemma 12.** *Suppose $f_0$ is $L$-smooth and Assumption 1 holds. Then for all $x \in \mathbb{R}^d$ the following estimate is valid:*

$$-\gamma\left\langle \nabla f_0(x), \frac{1}{G}\sum_{i\in\mathcal{G}}\nabla f_i(x)\right\rangle \quad \leqslant \quad -\frac{\gamma}{2}\|\nabla f(x)\|^2 + \gamma\cdot\zeta(N) + \frac{3\gamma}{2}\left(\delta_1 + \delta_2\|\nabla f(x)\|^2\right).$$

*Proof.* We commence by examining the difference $\nabla f(x) - \nabla f_0(x)$:

$$-\gamma\left\langle \nabla f_0(x), \frac{1}{G}\sum_{i\in\mathcal{G}}\nabla f_i(x)\right\rangle \quad = \quad \gamma\left\langle \nabla f(x) - \nabla f_0(x), \frac{1}{G}\sum_{i\in\mathcal{G}}\nabla f_i(x)\right\rangle - \gamma\left\langle \nabla f(x), \frac{1}{G}\sum_{i\in\mathcal{G}}\nabla f_i(x)\right\rangle.$$

Next, we continue with further manipulations on the first term:

$$\gamma \left\langle \nabla f(x) - \nabla f_0(x), \frac{1}{G} \sum_{i \in \mathcal{G}} \nabla f_i(x) \right\rangle \quad \leq \quad \frac{\gamma}{2} \|\nabla f(x) - \nabla f_0(x)\|^2 + \frac{\gamma}{2} \left\| \frac{1}{G} \sum_{i \in \mathcal{G}} \nabla f_1(x) \right\|^2$$

$$\leq \quad \frac{\gamma}{2} \left( \|\nabla f(x) - \nabla f_1(x)\|^2 + \|\nabla f_1(x) - \nabla f_0(x)\|^2 \right)$$

$$+ \frac{\gamma}{2} \left\| \frac{1}{G} \sum_{i \in \mathcal{G}} \nabla f_i(x) \right\|^2$$

$$\overset{\text{(Lemma 11)}}{\leqslant} \quad \gamma \left( \zeta(N) + \delta_1 + \delta_2 \|\nabla f(x)\|^2 \right) + \frac{\gamma}{2} \left\| \frac{1}{G} \sum_{i \in \mathcal{G}} \nabla f_i(x) \right\|^2,$$

and with the second term,

$$-\gamma \left\langle \nabla f(x), \frac{1}{G} \sum_{i \in \mathcal{G}} \nabla f_i(x) \right\rangle \quad = \quad -\frac{\gamma}{2} \|\nabla f(x)\|^2 - \frac{\gamma}{2} \left\| \frac{1}{G} \sum_{i \in \mathcal{G}} \nabla f_i(x) \right\|^2$$

$$+ \frac{\gamma}{2} \left\| \frac{1}{G} \sum_{i \in \mathcal{G}} (\nabla f_i(x) - \nabla f(x)) \right\|^2$$

$$\leq \quad -\frac{\gamma}{2} \|\nabla f(x)\|^2 - \frac{\gamma}{2} \left\| \frac{1}{G} \sum_{i \in \mathcal{G}} \nabla f_i(x) \right\|^2$$

$$+ \frac{\gamma}{2G} \sum_{i \in \mathcal{G}} \|(\nabla f_i(x) - \nabla f(x))\|^2$$

$$\leq \quad -\frac{\gamma}{2} \|\nabla f(x)\|^2 - \frac{\gamma}{2} \left\| \frac{1}{G} \sum_{i \in \mathcal{G}} \nabla f_i(x) \right\|^2$$

$$+ \frac{\gamma}{2} \left( \delta_1 + \delta_2 \|\nabla f(x)\|^2 \right).$$

$\square$

Further we establish the descent lemma, that is used throughout the proofs for `Grad-BANT` for convergence both in expectation and probability

**Lemma 13.** *Let $f_0$ be L-smooth convex, then the following holds for the iteration of `Grad-BANT`:*

$$f_0(x^{t+1}) \quad \leqslant \quad f_0(x^t) - \frac{\gamma \beta}{M} \left\langle \nabla f_0(x^t), \sum_{i \in \mathcal{G}} g_i^t \right\rangle + \frac{L \gamma^2 \beta}{2M} \sum_{i \in \mathcal{G}} \|g_i^t\|^2.$$

*Proof.* Actually, the update step of the Algorithm 1 is given by:

$$x^{t+1} = x^t - \gamma \sum_{i=1}^{n} \mathbb{I}_{[\theta_i^t > 0]} \omega_i^t g_i^t,$$

where $g_i^t = S^T S \nabla f_i(x^t)$ and $\sum_{i=1}^n \omega_i^t = 1$. Applying Jensen's inequality for the convex function $f_0$ and denoting $\overline{\omega}_i^t = \frac{[\theta_i^t]_0}{\sum_{j=1}^n [\theta_j^t]_0}$:

$$
\begin{aligned}
f_0(x^{t+1}) &= f_0\Big( \sum_{i=1}^n \omega_i^t \Big[ x^t - \gamma \mathbb{I}_{[\theta_i^t > 0]} g_i^t \Big] \Big) \\
&\leqslant \sum_{i=1}^n \omega_i^t f_0\Big( x^t - \gamma \mathbb{I}_{[\theta_i^t > 0]} g_i^t \Big) \\
&= \sum_{i \in \mathcal{B}} \omega_i^t f_0\Big( x^t - \gamma \mathbb{I}_{[\theta_i^t > 0]} g_i^t \Big) + \sum_{i \in \mathcal{G}} \omega_i^t f_0\Big( x^t - \gamma \mathbb{I}_{[\theta_i^t > 0]} g_i^t \Big) \\
&\leqslant \sum_{i \in \mathcal{B}} (1-\beta) \omega_i^{t-1} f_0(x^t) + \sum_{i \in \mathcal{B}} \beta \overline{\omega}_i^t f_0\Big( x^t - \gamma \mathbb{I}_{[\theta_i^t > 0]} g_i^t \Big) \\
&\quad + \sum_{i \in \mathcal{G}} (1-\beta) \omega_i^{t-1} f_0\big( x^t \big) + \sum_{i \in \mathcal{G}} \beta \overline{\omega}_i^t f_0\Big( x^t - \gamma \mathbb{I}_{[\theta_i^t > 0]} g_i^t \Big) \\
&= (1-\beta) f_0(x^t) + \sum_{i \in \mathcal{B}} \beta \overline{\omega}_i^t f_0\Big( x^t - \gamma \mathbb{I}_{[\theta_i^t > 0]} g_i^t \Big) + \sum_{i \in \mathcal{G}} \beta \overline{\omega}_i^t f_0\Big( x^t - \gamma \mathbb{I}_{[\theta_i^t > 0]} g_i^t \Big).
\end{aligned}
$$

In the inequality above, we make an estimation $f_0\Big( x^t - \gamma \mathbb{I}_{[\theta_i^t > 0]} g_i^t \Big) \leqslant f_0\big( x^t \big)$, since the indicator guarantees us that we do not increase the trial function $f_0$ by performing a step. By eliminating the weights $\omega_i^{t-1}$ accumulated from past iterations, we can rearrange the coefficients between Byzantine and honest workers in such a way that honest workers have higher weights. To achieve this, we sort the honest workers by increasing values of $f_0$ and assign them coefficients $\omega_i$ in decreasing order. This permutation ensures that honest workers have higher weights and Byzantine workers have lower weights. This operation is valid because if $\overline{\omega}$ for some Byzantine worker is higher than for a honest worker, then this Byzantine has a greater influence on $f_0$, and changing the weights would worsen the overall influence of these two workers. Therefore, with new weights $\{\widetilde{\omega}_i^t\}_{i=1}^n$:

$$
\begin{aligned}
f_0(x^{t+1}) &\leqslant (1-\beta) f_0(x^t) + \sum_{i \in \mathcal{B}} \beta \widetilde{\omega}_i^t f_0\Big( x^t - \gamma \mathbb{I}_{[\theta_i^t > 0]} g_i^t \Big) \\
&\quad + \sum_{i \in \mathcal{G}} \beta \widetilde{\omega}_i^t f_0\Big( x^t - \gamma \mathbb{I}_{[\theta_i^t > 0]} g_i^t \Big) \\
&\leqslant (1-\beta) f_0(x^t) + \sum_{i \in \mathcal{B}} \beta \widetilde{\omega}_i^t f_0(x^t) + \sum_{i \in \mathcal{G}} \beta \widetilde{\omega}_i^t f_0\Big( x^t - \gamma \mathbb{I}_{[\theta_i^t > 0]} g_i^t \Big) \\
&= f_0(x^t) + (1-\beta) \big[ f_0(x^t) - f_0(x^t) \big] + \sum_{i \in \mathcal{B}} \beta \widetilde{\omega}_i^t \big[ f_0(x^t) - f_0(x^t) \big] \\
&\quad + \sum_{i \in \mathcal{G}} \beta \widetilde{\omega}_i^t \Big[ f_0\Big( x^t - \gamma \mathbb{I}_{[\theta_i^t > 0]} g_i^t \Big) - f_0(x^t) \Big].
\end{aligned}
$$

Let us assign the coefficient $\frac{1}{n}$ to all honest workers. This procedure is also valid. We sorted the weights and honest workers now have the greatest weights, thus, the sum of the coefficients of honest workers is at least $\frac{G}{M}$. Moreover, the honest workers with the stronger influence have the greater weights which allows to equalize the total weight $\frac{G}{M}$ between all $G$ workers. Thus, we get

$$
f_0(x^{t+1}) \leqslant f_0(x^t) + \frac{\beta}{n} \sum_{i \in \mathcal{G}} \Big[ f_0\Big( x^t - \gamma \mathbb{I}_{[\theta_i^t > 0]} g_i^t \Big) - f_0(x^t) \Big].
$$

Now we can remove the indicator function because if $g_i^t$ minimizes the trial function, the indicator equals 1. If $g_i^t$ maximizes the trial function, the indicator excludes this gradient. However, we still

account for it and maximize the trial function, thus:

$$
\begin{aligned}
f_0(x^{t+1}) &\leqslant f_0(x^t) + \frac{\beta}{M} \sum_{i \in \mathcal{G}} \left[ f_0\left(x^t - \gamma g_i^t\right) - f_0(x^t) \right] \\
&\leqslant f_0(x^t) + \frac{\beta}{M} \sum_{i \in \mathcal{G}} \left[ f_0(x^t) - \gamma \left\langle \nabla f_0(x^t), g_i^t \right\rangle + \frac{L\gamma^2}{2} \|g_i^t\|^2 - f_0(x^t) \right] \\
&= f_0(x^t) - \frac{\gamma\beta}{M} \left\langle \nabla f_0(x^t), \sum_{i \in \mathcal{G}} g_i^t \right\rangle + \frac{L\gamma^2\beta}{2M} \sum_{i \in \mathcal{G}} \|g_i^t\|^2.
\end{aligned}
$$

$\square$

We are now ready to present final results for the convex case. Convergence in expectation is derived as follows:

**Theorem 4.** *Let $f_0$ be L-smooth, convex, and satisfy 1 with $\delta_2 < \frac{1}{3}$, then, after $T$ iterations of Algorithm 1 with $\gamma \leqslant \frac{k(1-3\delta_2)}{4Ln(1+\delta_2)}$, the following holds:*

$$
\frac{1}{T} \sum_{t=0}^{T-1} \mathbb{E}\|\nabla f(x^t)\|^2 \leqslant \frac{f_0(x^0) - f_0(\hat{x}^*)}{\gamma T} \cdot \frac{4M}{\beta G} + 7\delta_1 + 4\zeta(N).
$$

*Proof.* According to the Lemma equation 13:

$$
\begin{aligned}
f_0(x^{t+1}) &\leqslant f_0(x^t) - \gamma\beta \left\langle \nabla f_0(x^t), \frac{1}{M} \sum_{i \in \mathcal{G}} g_i^t \right\rangle + \frac{L\gamma^2\beta}{2M} \sum_{i \in \mathcal{G}} \|g_i^t\|^2 \\
&= f_0(x^t) - \gamma\beta \left\langle \nabla f_0(x^t), \frac{1}{M} \sum_{i \in \mathcal{G}} S^T S \nabla f_i(x^t) \right\rangle + \frac{L\gamma^2\beta}{2M} \sum_{i \in \mathcal{G}} \left\| S^T S \nabla f_i(x^t) \right\|^2 \\
&= f_0(x^t) - \gamma\beta \cdot \frac{G}{M} \left\langle \nabla f_0(x^t), \frac{1}{G} \sum_{i \in \mathcal{G}} S^T S \nabla f_i(x^t) \right\rangle + \frac{L\gamma^2\beta}{2M} \sum_{i \in \mathcal{G}} \left\| S^T S \nabla f_i(x^t) \right\|^2
\end{aligned}
$$

Taking the expectation of both sides of the inequality:

$$
\begin{aligned}
\mathbb{E}f_0(x^{t+1}) &\leqslant \mathbb{E}f_0(x^t) - \gamma\beta \cdot \frac{G}{M} \left\langle \nabla f_0(x^t), \frac{1}{G} \sum_{i \in \mathcal{G}} \nabla f_i(x^t) \right\rangle \\
&\quad + \frac{L\gamma^2\beta 2n}{2Mk} \sum_{i \in \mathcal{G}} \left\| \nabla f_i(x^t) \right\|^2 \\
&\overset{\text{(Lemma 12)}}{\leqslant} \mathbb{E}f_0(x^t) + \frac{\gamma\beta G}{M}\zeta(N) - \frac{\gamma\beta G}{2M} \left\| \nabla f(x^t) \right\|^2 \\
&\quad + \frac{3\gamma\beta G}{2M} \left( \delta_1 + \delta_2\|\nabla f(x^t)\|^2 \right) + \frac{L\gamma^2\beta 2n}{2Mk} \sum_{i \in \mathcal{G}} \left\| \nabla f_i(x^t) \right\|^2 \\
&\leqslant \mathbb{E}f_0(x^t) + \frac{\gamma\beta G}{M}\zeta(N) - \frac{\gamma\beta G}{2M} \left\| \nabla f(x^t) \right\|^2 + \frac{3\gamma\beta G}{2M} \left( \delta_1 + \delta_2\|\nabla f(x^t)\|^2 \right) \\
&\quad + \frac{2L\gamma^2\beta 2n}{2Mk} \sum_{i \in \mathcal{G}} \mathbb{E} \left\| \nabla f(x^t) - f_i(x^t) \right\|^2 + \frac{2L\gamma^2\beta 2nG}{2Mk} \left\| \nabla f(x^t) \right\|^2 \\
&\leqslant \mathbb{E}f_0(x^t) + \frac{\gamma\beta G}{n}\zeta(N) - \frac{\gamma\beta G}{2M} \left\| \nabla f(x^t) \right\|^2 + \frac{3\gamma\beta G}{2M} \left( \delta_1 + \delta_2\|\nabla f(x^t)\|^2 \right) \\
&\quad + \frac{2L\gamma^2\beta 2nG}{2Mk} \left( \delta_1 + \delta_2\|\nabla f(x^t)\|^2 \right) + \frac{2L\gamma^2\beta 2nG}{2Mk} \left\| \nabla f(x^t) \right\|^2 \\
&\leqslant \mathbb{E}[f_0(x^t)] - \frac{\gamma\beta G}{2M} \left[ 1 - \frac{2L\gamma 2n}{k} - \left( 3 + \frac{2L\gamma 2n}{k} \right) \delta_2 \right] \|\nabla f(x^t)\|^2 \\
&\quad + \frac{\gamma\beta G}{2M} \left( 3 + \frac{2L\gamma 2n}{k} \right) \delta_1 + \frac{\gamma\beta G}{M}\zeta(N).
\end{aligned}
$$

We first fix $\delta_2 < \frac{1}{3}$. Then by choosing $\gamma \leqslant \frac{k(1-3\delta_2)}{4LM(1+\delta_2)}$ and summing over the iterations, we get the bound:

$$\frac{1}{T}\sum_{t=0}^{T-1}\mathbb{E}\|\nabla f(x^t)\|^2 \;\;\leqslant\;\; \frac{f_0(x^0) - f_0(x_*)}{\gamma T} \cdot \frac{4M}{\beta G} + 3\delta_1 + 4\zeta(N).$$

$\square$

However, the main result is the derivation the convergence with high probability

**Theorem 5.** *(Theorem 1 from Main Part) Let $f_0$ be $L$-smooth, convex, and satisfy 1 with $\delta_2 < \frac{1}{16}$, then, after $T$ iterations of Algorithm 1 with $\gamma \leqslant \frac{1-3\delta_2-5\varepsilon-5\varepsilon\delta_2}{2L(1+\varepsilon)^2(1+\delta_2)}$, the following holds:*

$$\frac{1}{T}\sum_{t=0}^{T-1}\|\nabla f(x^t)\|^2 \leq \frac{f_0(x^0) - f_0(x_*)}{\gamma T}\frac{4M}{\beta G} + 11\delta_1 + 10\zeta(N),$$

*with probability $1 - \delta$, where the reduced dimension $k = \Theta\left(\frac{\log 2TM/\delta}{\varepsilon^2}\right)$.*

*Proof.*

$$-\left\langle \nabla f_0(x^t), \frac{1}{G}\sum_{i\in\mathcal{G}} S^T S \nabla f_i(x^t) \right\rangle = -\left\langle S\nabla f_0(x^t), \frac{1}{G}\sum_{i\in\mathcal{G}} S\nabla f_i(x^t) \right\rangle$$

$$\leq -\left\langle \nabla f_0(x^t), \frac{1}{G}\sum_{i\in\mathcal{G}} \nabla f_i(x^t) \right\rangle + \varepsilon\left\|\nabla f_0(x^t)\right\| \cdot \left\|\frac{1}{G}\sum_{i\in\mathcal{G}} \nabla f_i(x^t)\right\|$$

$$\leq -\left\langle \nabla f_0(x^t), \frac{1}{G}\sum_{i\in\mathcal{G}} \nabla f_i(x^t) \right\rangle + \frac{\varepsilon}{2}\|\nabla f_0(x^t)\|^2 + \frac{\varepsilon}{2}\left\|\frac{1}{G}\sum_{i\in\mathcal{G}} \nabla f_i(x^t)\right\|^2$$

$$\leq -\left\langle \nabla f_0(x^t), \frac{1}{G}\sum_{i\in\mathcal{G}} \nabla f_i(x^t) \right\rangle + \frac{\varepsilon}{2}\|\nabla f_0(x^t)\|^2 + \frac{\varepsilon}{2}\frac{1}{G}\sum_{i\in\mathcal{G}}\|\nabla f_i(x^t)\|^2$$

$$\leq -\left\langle \nabla f_0(x^t), \frac{1}{G}\sum_{i\in\mathcal{G}} \nabla f_i(x^t) \right\rangle + \frac{3\varepsilon}{2}\|\nabla f_0(x^t) - \nabla f_1(x^t)\|^2 + \frac{3\varepsilon}{2}\|\nabla f_1(x^t) - \nabla f(x^t)\|$$

$$+\;\; \frac{3\varepsilon}{2}\|\nabla f(x^t)\|^2 + \frac{\varepsilon}{2}\frac{1}{G}\sum_{i\in\mathcal{G}}\|\nabla f_i(x^t)\|^2$$

$$\leq -\left\langle \nabla f_0(x^t), \frac{1}{G}\sum_{i\in\mathcal{G}} \nabla f_i(x^t) \right\rangle + \frac{3\varepsilon}{2}\zeta(N) + \frac{3\varepsilon}{2}\left(\delta_1 + \delta_2\|\nabla f(x^t)\|^2\right)$$

$$+\;\; \frac{3\varepsilon}{2}\|\nabla f(x^t)\|^2 + \frac{\varepsilon}{2}\frac{1}{G}\sum_{i\in\mathcal{G}}\|\nabla f_i(x^t)\|^2$$

with probability $1 - 2\delta$.

Using Lemma 9 we obtain

$$
\begin{aligned}
f_0(x^{t+1}) &\leq f_0(x^t) - \gamma\beta \cdot \frac{G}{M} \left\langle \nabla f_0(x^t), \frac{1}{G} \sum_{i \in \mathcal{G}} S^T S \nabla f_i(x^t) \right\rangle + \frac{L\gamma^2\beta}{2M} \sum_{i \in \mathcal{G}} \left\| S^T S \nabla f_i(x^t) \right\|^2 \\
&\leq f_0(x^t) - \gamma\beta \cdot \frac{G}{M} \left\langle \nabla f_0(x^t), \frac{1}{G} \sum_{i \in \mathcal{G}} \nabla f_i(x^t) \right\rangle + \frac{\gamma\beta G}{M} \frac{3\varepsilon}{2} \zeta(N) \\
&\quad + \frac{\gamma\beta G}{M} \frac{3\varepsilon}{2} \left( \delta_1 + \delta_2 \|\nabla f(x^t)\|^2 \right) + \frac{\gamma\beta G}{M} \frac{3\varepsilon}{2} \|\nabla f(x^t)\|^2 \\
&\quad + \left( \frac{L\gamma^2\beta(1+\varepsilon)^2}{2M} + \frac{\gamma\beta G}{M} \frac{\varepsilon}{2G} \right) \sum_{i \in \mathcal{G}} \|\nabla f_i(x^t)\|^2 \\
&\leq f(x^t) - \frac{\gamma\beta G}{2M} \left( 1 - 5\varepsilon - 2L\gamma(1+\varepsilon)^2 - \left( 3 + 2L\gamma(1+\varepsilon)^2 + 5\varepsilon \right) \delta_2 \right) \|\nabla f(x^t)\|^2 \\
&\quad + \frac{\gamma\beta G}{2M} \left( 3 + 2L\gamma(1+\varepsilon)^2 + 2\varepsilon \right) \delta_1 + \frac{\gamma\beta G}{M} \left( 1 + \frac{3\varepsilon}{2} \right) \zeta(N)
\end{aligned}
$$

We first fix $\delta_2 \leq \frac{1}{16}$. Then, by choosing $\gamma \leq \frac{1 - 3\delta_2 - 5\varepsilon - 5\varepsilon\delta_2}{2L(1+\varepsilon)^2(1+\delta_2)}$ and summing over the iterations, we get the bound

$$
\frac{1}{T} \sum_{t=0}^{T-1} \|\nabla f(x^t)\|^2 \leq \frac{f_0(x^0) - f_0(x_*)}{\gamma T} \frac{4M}{\beta G} + 11\delta_1 + 10\zeta(N)
$$

with probability $1 - 2T\delta(M+1)$. Therefore, to satisfy probability $1 - \delta$ we need to take dimension $k = \Theta\left( \frac{\log 2TM/\delta}{\varepsilon^2} \right)$. $\qquad\square$

## C   PERSONALIZATION

Our analysis heavily relies on statements from (Hanzely et al., 2020). We derive necessary inequalities for the case with JL Transforms.

**Lemma 14.** *Define $x^*$ as following:*

$$x^* \;\;=\;\; \arg\min_z f_i(z) + \frac{\lambda}{2}\|z - y\|^2,$$

*Then, if $x$ is an $\Delta$-approximate solution to the subproblem in line 7 in Algorithm 2, then*

$$h_i(x) - f_i(x^*) - \frac{\lambda}{2}\|x^* - y\|^2 \le \Delta + \frac{\lambda\varepsilon}{2}\|x^* - y\|^2$$

*with probability $1 - 2\delta$.*

*Proof.* Define $x_S^*$

$$x_S^* \;\;=\;\; \arg\min_z f_i(z) + \frac{\lambda}{2}\|Sz - Sy\|^2.$$

Then, we have

$$h_i(x) - f_i(x^*) - \frac{\lambda}{2}\|x^* - y\|^2 \le \left( h_i(x) - f_i(x_S^*) - \frac{\lambda(1+\varepsilon)}{2}\|x_S^* - y\|^2 \right)$$
$$+ \left( f_i(x_S^*) + \frac{\lambda(1-\varepsilon)}{2}\|x_S^* - y\|^2 - f_i(x^*) - \frac{\lambda}{2}\|x^* - y\|^2 \right).$$

We can bound the first term:

$$h_i(x) - f_i(x_S^*) - \frac{\lambda(1+\varepsilon)}{2}\|x_S^* - y\|^2 \le h_i(x) - f_i(x_S^*) - \frac{\lambda}{2}\|Sx_S^* - Sy\|^2 \le \Delta,$$

where the inequality holds with probability $1 - \delta$. Focusing on the second term we obtain

$$f_i(x_S^*) + \frac{\lambda(1-\varepsilon)}{2}\|x_S^* - y\|^2 - f_i(x^*) - \frac{\lambda}{2}\|x^* - y\|^2$$
$$\le \;\; f_i(x_S^*) + \frac{\lambda}{2}\|Sx_S^* - Sy\|^2 - f_i(x^*) - \frac{\lambda}{2}\|x^* - y\|^2$$
$$\le \;\; f_i(x^*) + \frac{\lambda}{2}\|Sx^* - Sy\|^2 - f_i(x^*) - \frac{\lambda}{2}\|x^* - y\|^2$$
$$\le \;\; \frac{\lambda(1+\varepsilon)}{2}\|x^* - y\|^2 - \frac{\lambda}{2}\|x^* - y\|^2 = \frac{\lambda\varepsilon}{2}\|x^* - y\|^2.$$

The first and third inequalities hold with probability $1 - \delta$, and the second one due to the definition of minimum. $\qquad\square$

As $f_i \ge 0$ then

$$h_i(x) - f_i(x^*) - \frac{\lambda}{2}\|x^* - y\|^2 \le \Delta + \varepsilon\left( f_i(x^*) + \frac{\lambda}{2}\|x^* - y\| \right)$$

$$h_i(x) - (1+\varepsilon)\left( f_i(x^*) + \frac{\lambda}{2}\|x^* - y\|^2 \right) \le \Delta.$$

Therefore, $x$ is a $\Delta$-approximate solution to a $(1 + \varepsilon) \cdot \left( f_i(z) + \frac{\lambda}{2}\|z - y\|^2 \right)$. Consider $F' = (1+\varepsilon)F$. Optimal point has not changed, therefore by minimizing $F'$ we will minimize $F$ as well. In the literature the following result is obtained:

**Theorem 6** ((Hanzely et al., 2020)). *Suppose, that $f_i$ is $L$-smooth and $\mu$-strongly convex for all $i$. Let AGD with starting point $y_i^t$ be employed for $J_t \stackrel{def}{=} \sqrt{\frac{L+\lambda}{\mu+\lambda}} \log\left( 1152L\lambda M^2 \left( 2\frac{\lambda}{\mu} + 1 \right)^2 \mu^{-2} \right) + 4\sqrt{\frac{\mu(L+\lambda)}{\lambda(\mu+\lambda)}} t$ iterations to approximately solve subproblem at iteration $t$ and $T$ iterations overall. Then, we have*

$$f(x^T) - f_* \le 8\left( 1 - \sqrt{\frac{\mu}{\lambda}} \right)^T \left( F(x^0) - F_* \right)$$

Since every $f_i$ was $L$-smooth and $\mu$-strongly convex, $f_i'$ is $(1 + \varepsilon)L$-smooth and $(1 + \varepsilon)\mu$-strongly convex. As the new problem is solved with new personalization level $(1+\varepsilon)\lambda$, number of conducted local steps by AGD does not change. Therefore, we have

**Theorem 7.** *(Theorem 2 from Main Part) Let functions $f_i$ be $L$-smooth, $\mu$-strongly convex and $f_i \geq 0$ for all $i$. Let AGD with starting point $y_i^t$ be employed for $J_t \overset{def}{=} \sqrt{\frac{L+\lambda}{\mu+\lambda}} \log\left(1152 L\lambda M^2 \left(2\frac{\lambda}{\mu} + 1\right)^2 \mu^{-2}\right) + 4\sqrt{\frac{\mu(L+\lambda)}{\lambda(\mu+\lambda)}} t$ iterations to approximately solve sub-problem at iteration $t$ and $T$ iterations overall. Then, we have*

$$f(x^T) - f_* = \mathcal{O}\left(\left(1 - \sqrt{\frac{\mu}{\lambda}}\right)^T \left(f(x^0) - f_*\right)\right)$$

*with probability $1 - \delta$, where*

$$k \geq \frac{4\log\frac{2MT}{\delta}}{\varepsilon^2/2 - \varepsilon^3/3}.$$

**Remark 1.** *Assumption of $f_i \geq 0$ is not crucial and restrictive, since one may consider $f_i + C$, where $C \geq -f_i(x_i^*)$, if the estimation of the minimum is known. Or, in most machine learning problems, $f_i^* \geq 0$, therefore no additional knowledge is needed.*

## D   ADMM

We consider ADMM with linear JL Transforms. First of all, we show, that applying these dimension reductions do not change feasibility set with high probability. Then, we demonstrate, that

**Lemma 15.** *If $S$ is a JL Matrix, then* $\ker S = \{0\}$ *with probability* $1 - \delta$

*Proof.* According to Definition 4 we have
$$\mathbb{P}\left[(1 - \varepsilon)\|x\|^2 \leq \|Sx\|^2 \leq (1 + \varepsilon)\|x\|^2\right] \geq 1 - \delta, \quad \forall x \in \mathbb{R}^n.$$
Therefore, $\forall x \neq 0$, then,
$$\|Sx\|^2 \geq (1 - \varepsilon)\|x\|^2 \geq 0$$
with probability at least $1 - \delta$, and $x \notin \ker S$.

Lastly, $0 \in \ker S$ due to the linearity. $\square$

**Lemma 16.** *Algorithm 4 is the ADMM method for the problem*
$$\min_{x,y} l(\sum z_i, b) + \sum_i r_i(x_i) \tag{10}$$
$$s.t. \quad SA_i x_i - S z_i = 0$$

*Proof.* According to section 8.3 from Boyd et al. (2011) Algorithm 4 can be rewritten as
$$x_i^{t+1} = \arg\min_{x_i} \left( r_i(x_i) + \frac{\rho}{2}\|SA_i x_i - S z_i^t + u_i^t\|^2 \right)$$
$$z^{t+1} = \arg\min_z \left( l\left( \sum_{i=1}^M z_i; b \right) + \sum_{i=1}^M \frac{\rho}{2}\|SA_i x_i^{t+1} - S z_i^t + u_i^t\|^2 \right)$$
$$u_i^{t+1} = u_i^t + SA_i x_i^{t+1} - S z_i^{t+1}.$$
This update scheme is equivalent to scaled ADMM version, applied to (10) $\square$

From the the general ADMM convergence Boyd et al. (2011) we have
$$l\left( \sum z_i^t, b \right) + \sum_i r_i(x_i^t) \to p_*,$$

**Lemma 17.** *With probability at least* $1 - \delta$ *problems (4) and (10) have the same optimal value.*

*Proof.* If pair $(x_i, z_i)$ is in feasible set of problem 10, then,
$$SA_i x_i - S z_i = S(A_i x_i - z_i) = 0.$$
According to Lemma 15, $\ker S = \{0\}$ with probability at least $1 - \delta$, hence $A_i x_i - z_i = 0$ and $(x_i, z_i)$ is in problem (4) feasible set. On the other way, $A_i x_i - z_i = 0$ implies $SA_i x_i - S z_i = 0$. Therefore, with probability $1 - \delta$ these sets coincide and
$$\inf \left\{ l\left( \sum_i z_i, b \right) + \sum_i r_i(x_i) \mid SA_i x_i - S z_i = 0 \right\} = \inf \left\{ l\left( \sum_i z_i, b \right) + \sum_i r_i(x_i) \mid A_i x_i - z_i = 0 \right\}.$$
$\square$

**Theorem 8.** *(Theorem 3 from Main Part)*

*Proof.* It remains to combine Lemma 17 and Theorem 4.1 from He & Yuan (2012).

$\square$

# E  ADDITIONAL NUMERICAL INFORMATION

## E.1  PERSONALIZATION

We provide additional comparison of APGD method on datasets `a9a`, `w8a` and `mushrooms`. All experiments are conducted on A100.

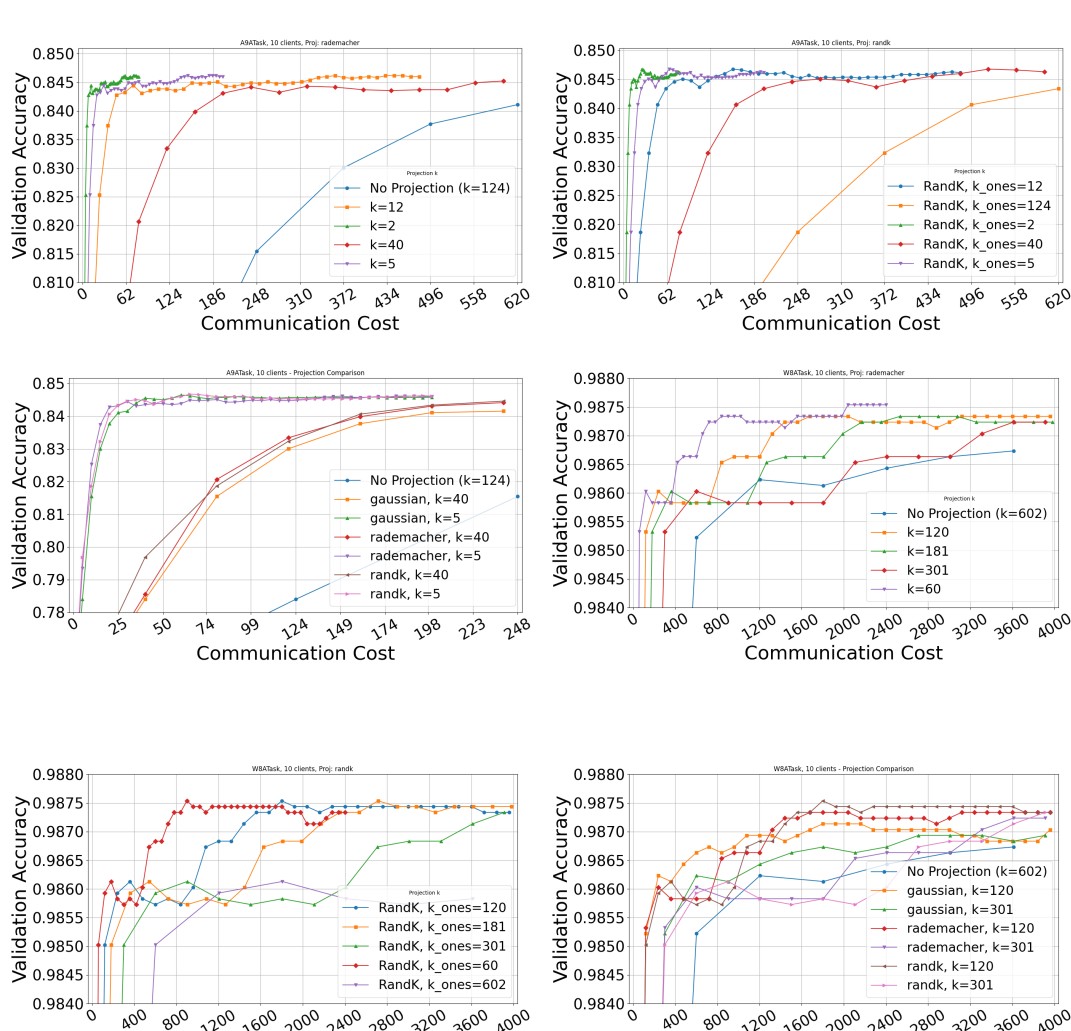

## E.2  GRAD-BANT IN DISTRIBUTED OPTIMIZATION

In the manuscript we also investigate the adversaries' influence over the convergence process. We consider such attacks, as Label Flipping and SignFlip. We define the number of Byzantine clients as a percentage of the total number of clients (Figure 1). We train `Grad-BANT` as well as existing methods: Zeno (Xie et al., 2019), Recess (Yan et al., 2023), FLTrust (Cao et al., 2020), Centered Clip (Karimireddy et al., 2021; Allouah et al., 2023), Safeguard (Allen-Zhu et al., 2020) and basic Adam (Kingma & Ba, 2014), which is not attack-robust.

Additionally, we compare `Grad-BANT` with `PowerSGD` Vogels et al. (2019), a well-established benchmark in distributed optimization, utilizing communication compressions. We study its behaviour on training `ResNet-18` on `CIFAR-10`.

At Figures 6a and 6b "Compression" stands for the number of reduced information - "Compression" 90% stands for 10 times dimension reduction, while 50% only for 2 times. Appears, that more aggressive reduction demonstrate better performance.

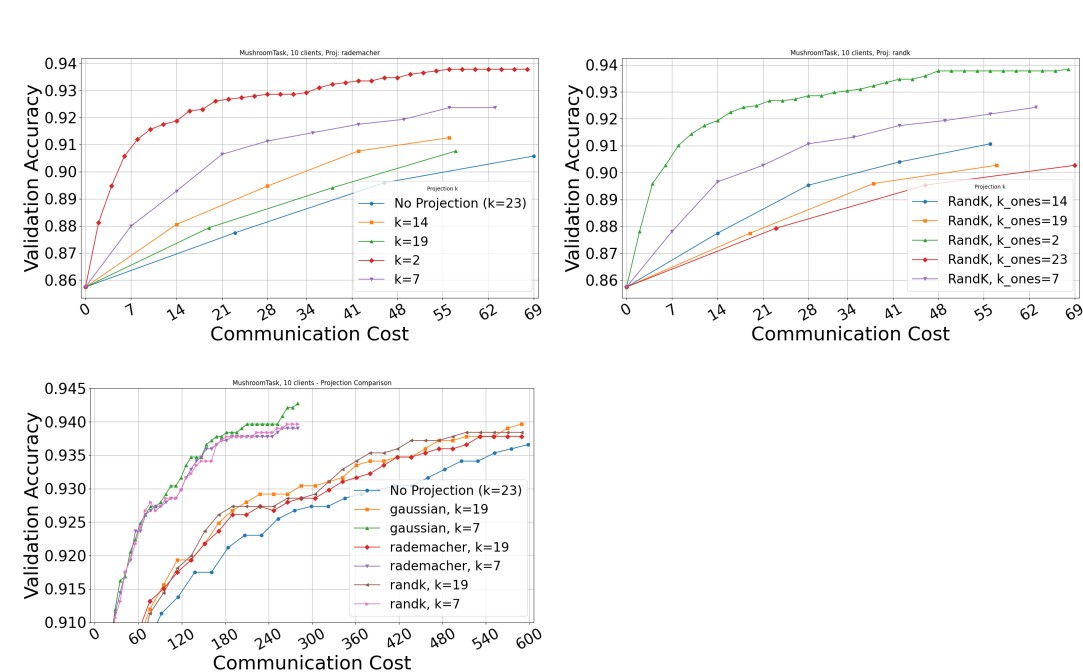

Figure 4: APGD perfomance on LibSVM datasets

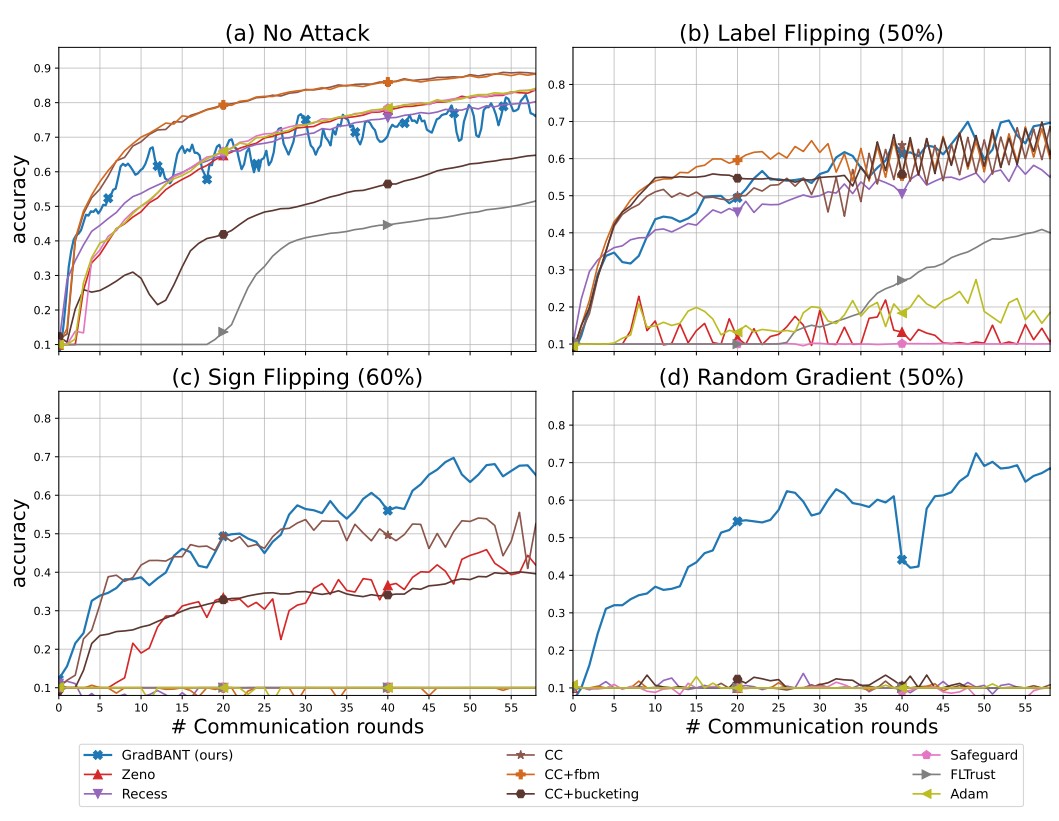

Figure 5: Byzantine Attacks

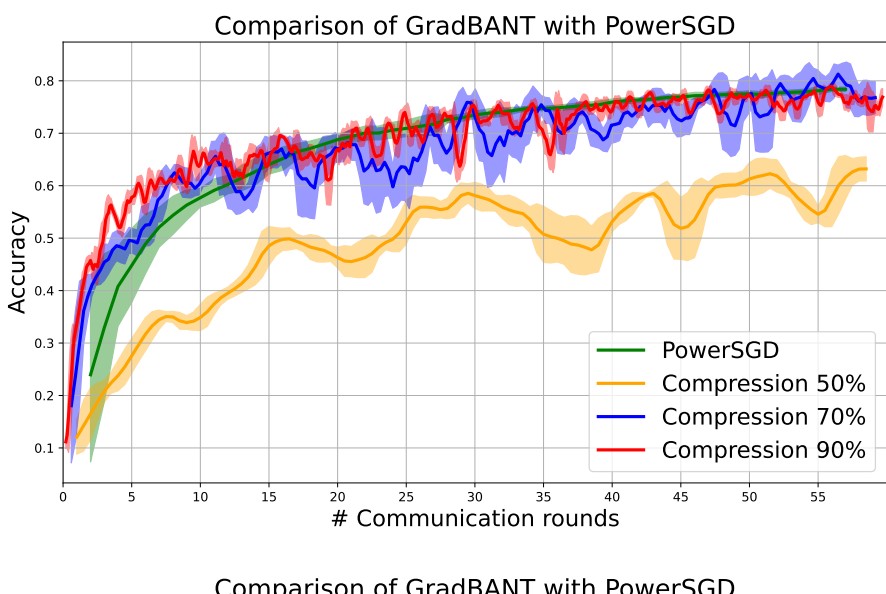

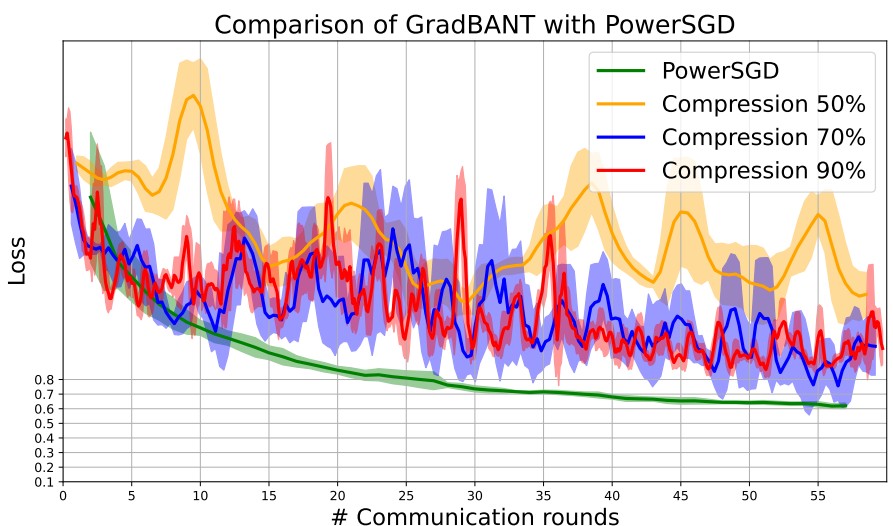

Figure 6: Comparison with `PowerSGD`

Further we compare different compression percentages and choice of random matrix influences the overall convergence. We analyze 70, 90 and 95 %. From Figure 7 it can be seen, that Randk is consistently worse, than our proposed schemes.

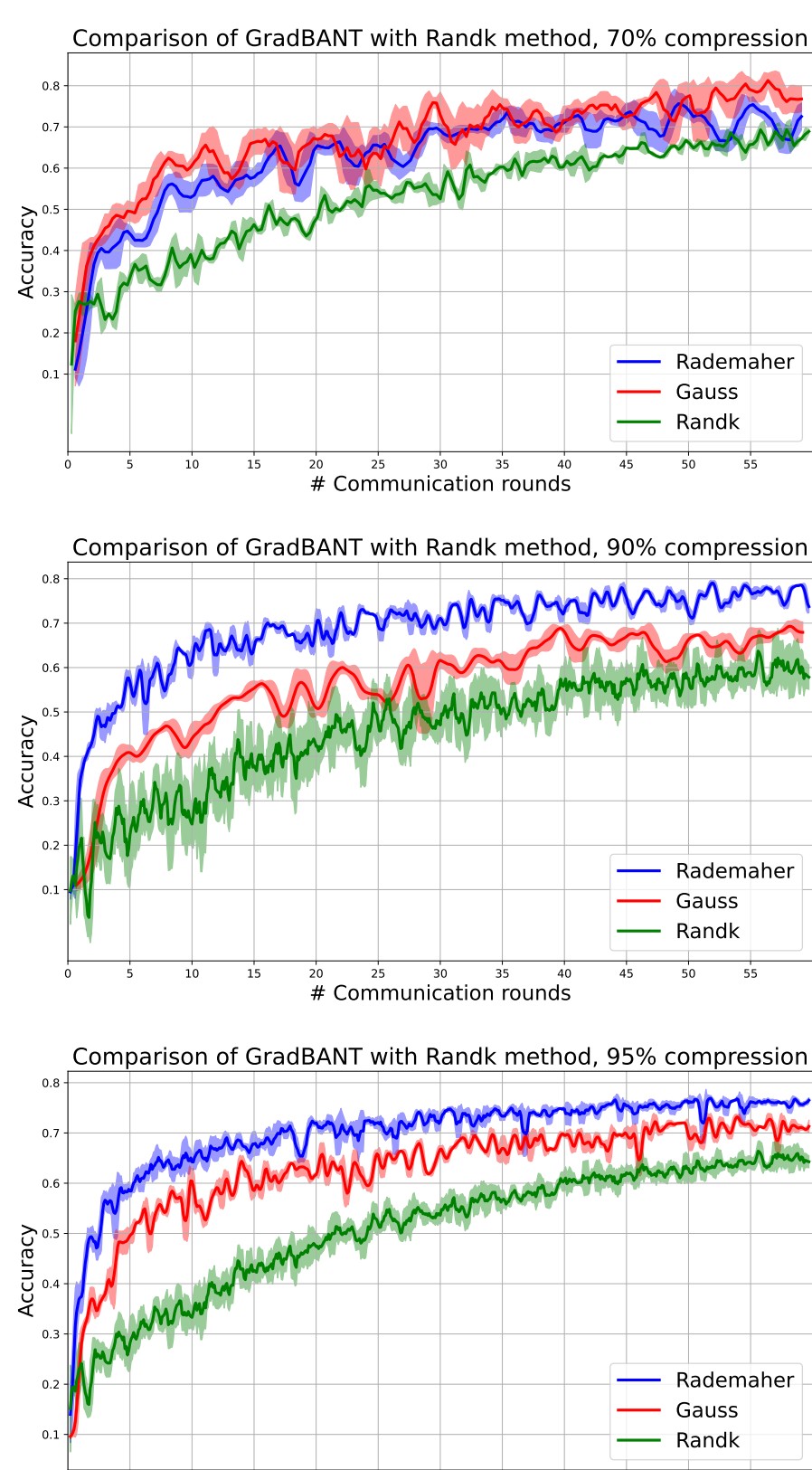

Figure 7: Varying compression levels.

# F   DECLARATION OF LLM USAGE

We employed Large Language Models to improve the clarity and style of the text.

