# OpenReview forum: "Johnson-Lindenstrauss Transforms in Distributed Optimization"
_ICLR.cc/2026/Conference — Submitted to ICLR 2026_

### Official Review · Reviewer_McCk · 2025-10-23

**Soundness:** 2
**Presentation:** 2
**Contribution:** 2
**Rating:** 4
**Confidence:** 2

**Summary:**

This paper integrates Johnson-Lindenstrauss (JL) Transforms into distributed optimization to mitigate communication bottlenecks and other issues in distributed optimization. The authors give comprehensive convergence analyses, as well as numerical experiments to support the theory.

**Strengths:**

1. The paper fills a critical gap by leveraging JL’s unique distance/scalar-product preservation.
2. The convergence analyses are comprehensive.

**Weaknesses:**

1. The designed experiments are relatively simple. Datasets such as a9a and w8a fail to fully demonstrate the algorithm performance. Additionally, CIFAR-10 is relatively small, so it is recommended to try larger datasets like CIFAR-100 or Tiny-ImageNet. Larger-scale experiments are required to fully verify the effectiveness of the proposed method.
2. The core focus of the paper is not clear enough. In the experiment section, the proposed method is compared with three types of methods: a. communication-efficient methods, b. personalized methods, and c. vertical FL.  However, each category lacks sufficient depth. It is suggested that the research scope be narrowed down to ensure a more in-depth analysis.
3. There is no ablation study related to critical parameters.
4. Line 431, Missing reference.

**Questions:**

Please see weaknesses

---

### Official Review · Reviewer_8yCp · 2025-10-30

**Soundness:** 3
**Presentation:** 2
**Contribution:** 2
**Rating:** 4
**Confidence:** 2

**Summary:**

This paper proposes to incorporate Johnson–Lindenstrauss (JL) transforms into distributed optimization frameworks, offering a unifying view of distance-preserving compression for diverse settings including Byzantine-robust training, personalized federated learning, and vertically partitioned ADMM. The idea is elegant and theoretically sound, as JL projections naturally preserve geometric structures that many existing compression operators destroy. The authors develop modified algorithms with provable high-probability convergence guarantees and provide extensive empirical results on both convex and nonconvex problems.

**Strengths:**

1. The use of JL transforms as communication compressors is novel and grounded in well-established random projection theory. The high-probability analysis extends beyond standard unbiased-compression assumptions, which makes the theoretical contribution nontrivial.
2. Applying the same principle to three distinct distributed setups—Byzantine-robust, personalized, and vertical FL—demonstrates generality. The paper’s structure is coherent, and the notation is consistent across sections.
3. Experiments are extensive and include diverse datasets and attacks; results consistently show that JL-based compression can outperform random sparsification and low-rank baselines while significantly reducing communication.

**Weaknesses:**

1. Although the conceptual introduction of JL transforms is interesting, the algorithmic modifications in the three settings are relatively straightforward adaptations. The originality lies more in applying an existing tool than in developing fundamentally new algorithms.
2. The paper does not thoroughly discuss the computational or memory cost of generating and applying large random matrices $S$ and  $S^\top$. For high-dimensional models, the matrix–vector multiplications may offset communication gains unless structured JL variants are used.
3. While experiments are numerous, details such as the exact compression ratio, random seed synchronization method, and matrix reuse strategy are briefly mentioned but not systematically evaluated. This makes reproduction and scalability assessment difficult.

**Questions:**

1. How do the authors handle the computational and memory overhead of multiplying by large random matrices $S$ and $S^\top$ in high-dimensional models ? Have the authors considered or tested structured JL transforms (such as sparse JL) to mitigate this cost?
2. Since federated clients may have highly non-IID data, how stable is JL-based compression under strong heterogeneity? Does the preservation of inner products in low-dimensional space still hold in practice when client updates are strongly biased?
3. The theoretical bounds depend on the JL dimension $k$, but the empirical section does not show how performance varies with $k$. Could the authors include sensitivity studies to clarify the minimal $k$ needed for reliable optimization and when performance starts to degrade?

---

### Official Review · Reviewer_Mpf9 · 2025-11-04

**Soundness:** 2
**Presentation:** 3
**Contribution:** 2
**Rating:** 4
**Confidence:** 4

**Summary:**

This paper presents a method of using Johnson-Lindenstrauss (JL) mappings to reduce communication while ensuring robustness in distributed learning and federated learning.  The paper primarily focused on proving that the JL mapping ensures convergence with high probability, so that the desired criterion is satisfied with high confidence, rather than on average.  This is based on connecting distributed optimization with JL Transforms' ability to maintain L2 distances between vectors.

The paper adapted the method for distributed learning for Byzantine-resilient optimization, building on the work of Grad-BANT.  In both the personalized federated learning setting and the vertical federated learning setting, JL is used to demonstrate that compressed communication provides a convergence guarantee.

The theory is well developed and clearly presented.

However, the experimental section is short and incomplete.  The JL mapping experiments are based on two types of stochastic projection matrices, Gaussian and Rademacher.  They do better than randomized sparsification on the mushroom dataset.  Training with ResNet-20 on CIFAR-10 is mentioned, but it is unclear which figure it refers to, nor is the learning setting specified.

**Strengths:**

The paper could serve as a good tutorial on demonstrating why JL is a suitable technique for compression in the distributed and federated learning setting.  It fills the knowledge gap by demonstrating the distributed optimization capabilities of JL Transforms, which maintain L2 distances between vectors. This ensures that the desired criterion can be satisfied with high confidence, rather than on average.

**Weaknesses:**

The experimental validation is weak.  It is challenging to assess the practical relevance of this method, as it primarily consists of an analysis of the convergence.

**Questions:**

none.

---

### Official Review · Reviewer_ecJS · 2025-11-08

**Soundness:** 1
**Presentation:** 2
**Contribution:** 1
**Rating:** 2
**Confidence:** 4

**Summary:**

The paper studies distributed optimization with data compression techniques. Specifically, the authors study how Johnson–Lindenstrauss (JL) mappings can be used for data compression in distributed optimization.

JL mappings are random mappings that preserve the $\ell_2$ norms between vectors when compressed, with high probability.

If JL mappings are linear, then they also ensure that inner products are preserved after compression, with high probability.

The paper considers three problems:

1. Byzantine-robust optimization where some workers can send adversarial gradients and the optimization process needs to be robust. This is done using a trusted device as a reference.

2. Personalized federated learning where each worker has their own data and objectives but they want the model parameters to still be similar across devices.

3. Vertical federated learning where the columns or features of the data are spread across the devices.

First, the authors show that JL transforms induce a $JL_s$ operator that is unbiased. The authors claim the $JL_s$ operator also preserves norms, which other compressors like Rand-$k$ do not.

For all three cases, inspired by existing algorithms in the non-compressed setting, the authors develop algorithms with JL-transform compression.

1. In the Byzantine case, the authors develop Grad-BANT that sends projected gradients. Under assumptions on the bound of the attack, that is, assumptions on data similarity, the authors claim that it achieves $O(1/T)$ convergence with high probability.

2. In the personalized federated learning case, the authors use an accelerated gradient–style approach from Hanzely et al. (2020) and show they achieve the same convergence rate as the uncompressed case.

3. In the vertical FL case, the authors show ADMM can be adapted to get the same saddle-point residual convergence.

The authors also give experimental evidence for these methods.

**Strengths:**

The problem studied in the paper is definitely interesting.

JL Transforms have been shown to be useful in many applications including distributed optimization. Further characterizing how we can use JL transforms for more problems and achieve high probability results would be of interest to the community.

The authors have experimental results showing that the algorithms they developed do lead to good validation accuracy.

**Weaknesses:**

One of my biggest concerns with this paper is that I think it interprets the JL transform lemma to be much stronger than what it actually implies.

In Definition 4, the authors assume that there is a single stochastic mapping $h$ that can uniformly satisfy, for all $u,v$ pairs:

$\Pr\\big[(1-\varepsilon)\\|u-v\\|_2^2 \le \\|h(u)-h(v)\\|_2^2 \le (1+\varepsilon)\\|u-v\\|_2^2\big] \ge 1-\delta$

This is not true. The JL transform lemma results presented in Johnson et al. (1984) and Dasgupta and Gupta (2003) show that the uniform guarantee that the norms are preserved simultaneously between all points is only true for a finite set. The probabilistic guarantee, which holds with high probability, is only for a single point or a single pair, which is then used in a union-bound to get the result for the finite set.

Whereas the authors are assuming that the high-probability result holds for all $x$ simultaneously. In other words, the JL transform says that if we fix an $x$, then if we sample an $S$ (or $h$), with high probability the norm of $x$ will be preserved. It does not say that if we sample an $S$, then for all $x$, the norms will be preserved with high probability. Since $S$ is a $k \times n$ matrix with $k < n$, by definition it cannot have full column rank, thus it necessarily has a non-trivial null space. For any vector in the null space of $S$, the norm will not be preserved. Thus, no fixed $S$ can satisfy the JL lemma simultaneously or uniformly.

I believe this is a critical flaw, and this result is used to further prove lemmas, which I believe are also untrue as a consequence.

For example:

1. In the proof for Lemma 4, presented as Lemma 9 in the supplementary material, the authors claim that since
$(1-\varepsilon)\\|x\\|^2 \le \\|Sx\\|^2 \le (1+\varepsilon)\\|x\\|^2$
then the eigenvalues of $S^\top S$ are between $[1-\varepsilon, 1+\varepsilon]$.

This conclusion would be true only if $(1-\varepsilon)\\|x\\|^2 \le\\|Sx\\|^2 \le (1+\varepsilon)\\|x\\|^2$ was true for all unit-norm $x$ simultaneously, but the JL lemma only holds when $x$ is fixed.

In fact, since $S$ is a $k \times n$ matrix, the rank of $S^\top S$ is at most $k$, and since $k < n$, there are at least $n - k$ eigenvalues of $S^\top S$ equal to $0$, which are obviously not in $[1-\varepsilon, 1+\varepsilon]$.

Moreover, the Marchenko-Pastur distribution shows that in the Gaussian JL matrix, the largest eigenvalue of $S^\top S$ scales as $(1 + \sqrt{k/n})^2$, which is larger than $1 + \varepsilon$.

This result is used to prove the main result for the Byzantine case, making that result unreliable.

2. Similarly, Lemma 15 says if $S$ is a JL matrix, then with high probability $\ker S = \{0\}$. This is also untrue. Since $k < n$, $S$ cannot have full column rank, thus $\dim \ker S \ge n - k$, i.e., a non-trivial null space.

This result is used to prove the result for the vertical FL, making it unreliable.

I have not checked all the proofs, so I am not too confident if there are more such issues.


Moreover, the writing of the paper can be improved and is confusing at times, and it is missing references. For example:
1. In Eq. (4), how did we change $r_i(x_i)$ to $r_i(z_i)$? Here $x_i \in \mathbb{R}^{n_i}$ whereas $z_i \in \mathbb{R}^{m}$.
2. The updates of ADMM are explained without introducing or mentioning what $\rho$ is.
3. Missing reference in the Experiments section, it says “At Figure ??”.
4. Proof of Lemma 11 says “Theorem 5 from (?)”.

**Questions:**

1. Can the authors address my concern about the overly strong assumption in the JL lemma? Specifically, can they show how this assumption can be rectified or prove that their claims still hold under the standard JL guarantees?

2. Which results remain valid if statements such as Lemma 4 and Lemma 15 are incorrect?

---

### Meta-Review · Area_Chair_EHfn · 2025-12-22

**Summary:**

1. The technical results in this paper are incorrect.
2. It lacks novelty by directly appling JL transform to certain optimization problems

**Reviewer Concerns:**

There is no rebuttal

**Reviewer Scores:**

Reviewers will maintain their score

---

### Decision · Program_Chairs · 2026-01-26

Reject